# TASK RELATEDNESS-BASED GENERALIZATION BOUNDS FOR META LEARNING

**Jiechao Guan**
School of Information, Renmin University of China, Beijing, China
`2014200990@ruc.edu.cn`

**Zhiwu Lu** [*]
Gaoling School of Artificial Intelligence, Renmin University of China, Beijing, China
Beijing Key Laboratory of Big Data Management and Analysis Methods, Beijing, China
`luzhiwu@ruc.edu.cn`

## ABSTRACT

Supposing the $n$ training tasks and the new task are sampled from the same environment, traditional meta learning theory derives an error bound on the expected loss over the new task in terms of the empirical training loss, uniformly over the set of all hypothesis spaces. However, there is still little research on how the relatedness of these tasks can affect the full utilization of all $mn$ training data (with $m$ examples per task). In this paper, we propose to address this problem by defining a new notion of task relatedness according to the existence of the bijective transformation between two tasks. A novel generalization bound of $\mathcal{O}(\frac{1}{\sqrt{mn}})$ for meta learning is thus derived by exploiting the proposed task relatedness. Moreover, when investigating a special branch of meta learning that involves representation learning with deep neural networks, we establish spectrally-normalized bounds for both classification and regression problems. Finally, we demonstrate that the relatedness requirement between two tasks is satisfied when the sample space possesses the completeness and separability properties, validating the rationality and applicability of our proposed task-relatedness measure.

## 1 INTRODUCTION

By leveraging knowledge distilled from the training tasks[1], meta learning (Thrun & Pratt, 1998; Baxter, 2000) learns to perform well on a new but related task. One important branch of meta learning achieving great success in practical machine learning applications is representation learning (Krizhevsky et al., 2012; Bengio et al., 2013; He et al., 2016), where one first learns a shared feature extractor (e.g., deep neural networks) over the training tasks, and then learns a prediction function for the new task on the top of the features constructed from the extractor, within a few gradient steps (Finn et al., 2017; Li et al., 2017) or with a feedforward process (Snell et al., 2017; Sung et al., 2018) . If the feature extractor can capture the common information across tasks, it is possible that utilizing representation learning can help learners generalize well to a new task with much less data.

To build up a rigorous framework to support this intuition, the pioneering meta learning theory assumes that both the $n$ training tasks and the new task are i.i.d. generated from the same environment (Baxter, 2000). Then, analogous to the single task learning whose goal is to select from the hypothesis space $\mathcal{H}$ a hypothesis $h$ to achieve the minimal expected loss on the task, meta learning expects to choose from the hypothesis space family $\mathbb{H}$ a hypothesis space $\mathcal{H}(\in \mathbb{H})$ that contains a good solution to any task sampled from the environment. Under the PAC learning framework (Valiant, 1984; Vapnik, 1989), (Baxter, 2000) derives a uniform bound of $\mathcal{O}(\sqrt{\frac{C_1}{mn}} + \sqrt{\frac{C_2}{n}})$ on the expected loss of a hypothesis space $\mathcal{H}$ over the new task in the environment, according to the empirical loss of $\mathcal{H}$

---

[*]Corresponding author.
[1]In this work, a task represents a data distribution, or say a probability measure on the sample space.

over the $n$ training tasks, where $C_1, C_2$ are different logarithms of the covering number (Anthony & Bartlett, 2002) about $\mathbb{H}$. However, compared with the single task learning whose error bound $\mathcal{O}(\sqrt{\frac{C}{m}})$ of any $h \in \mathcal{H}$ can utilize all $m$ training samples, where $C$ represents certain complexity indicators such as the VC-dimension (Blumer et al., 1989) or the entropy (Pollard, 1984) of $\mathcal{H}$, Baxter's meta learning generalization bound of any $\mathcal{H} \in \mathbb{H}$ cannot fully utilize all $mn$ training samples (e.g., with a bound of $\mathcal{O}(\frac{1}{\sqrt{nm}})$, without extra terms of $\mathcal{O}(\frac{1}{\sqrt{n}})$ or $\mathcal{O}(\frac{1}{\sqrt{m}})$). Nevertheless, there is still little theoretical research on how the relatedness of these tasks can affect the full utilization of all $mn$ training data in meta learning under Baxter's proposed i.i.d. task environmetn framework.

In this paper, we propose to address this problem by studying the task relatedness and provide a new generalization bound. To achieve full utilization of all training sample, our motivation is that different tasks need to be 'related' enough such that samples from various measures can be assumed to be generated from an 'almost' identical distribution. The equivalence relation of different data distributions actually corresponds to the measure-preserving isomorphism of their induced measure spaces[2] (see Definition 7). Therefore in this paper, we define a new notion of task relatedness called *almost Π-relatedness* in Definition 3 according to the existence of a measure-preserving bijective transformation between two measure spaces associated with different tasks. A PAC-style generalization error bound that fully utilize all training data is thus provided in Theorem 3, by exploiting the proposed task relatedness. We further employ this task relatedness notation to analyze the representation learning with deep neural networks, and establish non-parameter-count-based spectrally-normalized bounds for both classification and regression problems under the meta learning framework. Finally, we demonstrate the rationality of our proposed task-relatedness notion from a theoretical standpoint.

Our main contributions are summarized as follows:

**(1)** We propose a new notion of task relatedness, called *almost Π-relatedness*, by exploring the existence of a bijective transformation between two tasks in the environment. A novel PAC-style generalization error bound of $\mathcal{O}(\sqrt{\frac{C}{mn}})$ is thus derived for general meta learning by exploiting the proposed task relatedness, where $C$ captures the logarithm of the covering number of a hypothesis space family. Such bound thus can fully utilize the whole $n * m$ training data in meta learning.

**(2)** For meta learning that involves representation learning, we bound the covering number in Contribution **(1)** with two covering numbers that are both defined over a single task, making our results suitable to be combined with recent works of deep neural network in the single task learning. In particular, we provide the spectrally-normalized bounds of $\mathcal{O}(\sqrt{\frac{C_1 + nC_2}{nm}})$ for classification and regression problems in meta learning, where $C_1, C_2$ are certain complexities that are dependent on the matrix norms, but independent of the total number of the parameters, in deep neural networks.

**(3)** We rigorously demonstrate that, any two tasks are almost Π-related if the sample space is a complete separable metric space, validating the applicability of our proposed task-relatedness measure.

## 2 RELATED WORK

**Meta Learning Theory**. The first theoretical analysis for meta learning is performed by (Baxter, 2000), which gives a bound on the expected loss on the unseen task of any hypothesis space in terms of the empirical training loss. Under this framework, (Pentina & Ben-David, 2015) studies the generalization error bound of the hypothesis space in the case of kernel learning. Although many follow-up works also assume that all tasks are sampled from the same environment, they bound different kinds of generalization error from various perspectives. One important branch is to bound the transfer risk over the new task of a deterministic procedure, such as the linear feature transformation algorithm (Maurer, 2009), and the Bayes algorithm in PAC-Bayes theory (Pentina & Lampert, 2014; Amit & Meir, 2018). Among them, (Chen et al., 2020) derive a bound of $\mathcal{O}(\frac{1}{\sqrt{n}})$ from the algorithm stability perspective, and (Pentina & Lampert, 2014) derives a bound of $\mathcal{O}(\frac{1}{\sqrt{n}} + \frac{1}{\sqrt{m}})$(we suppress the complexity factor in the numerator for concision, similarly hereinafter). Another branch aims to bound the excess risk of the task specific function returned by ERM algorithm over the unseen task. They study meta learning theory from different views, such as multitask representation learning

---

[2]A measure space is a triple (Z,$\mathcal{A}$, P), where Z is the set, $\mathcal{A}$ is the $\sigma$-algebra on Z, P is a measure on $\mathcal{A}$.

(Maurer et al., 2016) and online learning (Khodak et al., 2019). Among them, (Du et al., 2021) and (Tripuraneni et al., 2020) escape from Baxter's proposed task environment setting, by defining task similarity or task diversity notion, obtaining similar excess risk bounds of $\mathcal{O}(\frac{1}{\sqrt{mn}} + \frac{1}{\sqrt{m}})$ in the case of high-dimensional transfer learning. Our work follows Baxter's proposed i.i.d. task environment framework, and gives an improved bound on the *generalization error of the hypothesis space*. The detailed comparisons between our results and that in (Baxter, 2000) are presented in Appendix C.1.

**Task Relatedness**. Another related work (Ben-David & Schuller, 2003) also proposes a task-relatedness concept, by defining the bijective transformation $\pi$ on the input space $X$. But in our work, the bijective transformation $\pi$ is imposed on the sample space $Z = X \times Y$ (where $Y$ is the output space). It is not difficult to see that the task relatedness considered in (Ben-David & Schuller, 2003) is actually a special case of our defined task-relatedness measure. We further validate the rationality of our proposed notion by revealing the existence of the bijective transformation $\pi$ on $Z$ when $Z$ is a complete separable metric space, making our results more applicable. The differences between our proposed task-relatedness notion and that in (Ben-David & Schuller, 2003) can be found in Appendix C.2. Besides, (Khodak et al., 2019) and (Du et al., 2021) also consider task-relatedness notions such as parameter-closeness or sharing a low-rank subspace for meta learning. To distinguish the notions proposed in this paper with others is truly one of our future directions.

## 3 PRELIMINARY

In this paper, we use uppercase letters (e.g., $Z$) to represent different spaces. The boldface $\mathbf{z}$ represents a matrix, with $\mathbf{z}_i$ as its $i$-th row and $\mathbf{z}_{:j}$ its $j$-th column. $\vec{z}$ denotes a vector. All detailed proofs of our theoretical results are deferred to the Appendix B for reader's benefits.

### 3.1 META LEARNING

The formulation of meta learning problem can be summarized as follows. We are given a sample space $Z = X \times Y$, where $X$ is an input space and $Y$ an output space. In this paper, we assume that $Z$ is a complete separable metric space. A loss function is defined as $l : Y \times Y \to \mathbb{R}$, and we assume that $l$ has the range $[0, 1]$, or equivalently, with rescaling technique, we assume that $l$ is bounded. An environment $(\mathcal{P}, Q)$ is a two-tuple, where $\mathcal{P}$ is the set of all probability measures\distributions\tasks on $X \times Y$ and $Q$ is a probability measure on $\mathcal{P}$. A hypothesis space family is formulated as $\mathbb{H} = \{\mathcal{H}\}$, where each $\mathcal{H} \in \mathbb{H}$ is a set of hypotheses\functions $h : X \to Y$. In single task learning, the learner needs to choose an optimal hypothesis $h^* \in \mathcal{H}$ to minimize the expected loss of $h$ over a probability measure $P(\in \mathcal{P})$ on the product space $X \times Y$: $er_P(h) = \int_{X \times Y} l(h(x), y) \mathrm{d}P(x, y)$. Similarly, the goal of meta learning is to find a hypothesis space $\mathcal{H}^* \in \mathbb{H}$ which contains a good solution to any task sampled from the environment, and minimize the following expected loss of $\mathcal{H}$ over a measure $Q$ on $\mathcal{P}$ (Baxter, 2000, Eq.(6)): $er_Q(\mathcal{H}) = \int_{\mathcal{P}} \inf_{h \in \mathcal{H}} er_P(h) \mathrm{d}Q(P)$. In practice, it is hard to minimize $er_Q(\mathcal{H})$ directly since we do not know the exact distribution of the environment measure $Q$. We can only capture the information about the environment $(\mathcal{P}, Q)$ by observing the training data $\mathbf{z}$ generated from the $n$ training tasks $P_j(j = 1, ..., n)$ that are i.i.d. sampled from the environment measure $Q$. Formally, this is achieved in the following manner: (1) Sample $n$ times from $\mathcal{P}$ according to $Q$ to generate (i.i.d.) probability measures $P_1, ..., P_n$. (2) Sample $m$ times from $X \times Y$ according to $P_j$ to generate $\{(x_{1j}, y_{1j}), ..., (x_{mj}, y_{mj})\}$ $(1 \leq j \leq n)$. (3) Denote $z_{ij} = (x_{ij}, y_{ij}) \in Z$, and an $(m, n)$-sample will be generated, denoted by $\mathbf{z}$ and written as a matrix $\mathbf{z} = (z_{ij})_{m \times n}$. We then choose to minimize the empirical loss $\hat{er}_{\mathbf{z}}(\mathcal{H})$ over the training data $\mathbf{z}$ for meta learning, which is defined as $\hat{er}_{\mathbf{z}}(\mathcal{H}) = \frac{1}{n} \sum_{j=1}^n \inf_{h \in \mathcal{H}} \hat{er}_{\mathbf{z}_{:j}}(h)$, where $\mathbf{z}_{:j}$ is the $j$-th column of matrix $\mathbf{z}$. Let $\mathbf{P} = (P_1, ..., P_n)$. We also consider the following loss $\hat{er}_{\mathbf{P}}(\mathcal{H})$ as the empirical estimate of the expected loss $er_Q(\mathcal{H})$: $\hat{er}_{\mathbf{P}}(\mathcal{H}) = \frac{1}{n} \sum_{j=1}^n \inf_{h \in \mathcal{H}} er_{P_j}(h)$.

**Definition 1** *(Baxter, 2000) For any hypothesis $h : X \to Y$, define $h_l$ as the composition of loss function and hypothesis: $h_l : X \times Y \to [0, 1]$ by $h_l(x, y) = l(h(x), y)$. For any hypothesis space $\mathcal{H} \in \mathbb{H}$, define $\mathcal{H}_l = \{h_l : h \in \mathcal{H}\}$ as the function space on $Z$. For any sequence of $n$ hypothesises $(h_1, ..., h_n)$, define $(h_1, ..., h_n)_l : (X \times Y)^n \to [0, 1]$ by $(h_1, ..., h_n)_l(x_1, y_1, ..., x_n, y_n) = 1/n \sum_{i=1}^n l(h_i(x_i), y_i)$. We will use $\mathbf{h}_l$ to denote $(h_1, ..., h_n)_l$. $\forall \mathcal{H} \in \mathbb{H}$, define $\mathcal{H}_l^n = \{(h_1, ..., h_n)_l : h_1, ..., h_n \in \mathcal{H}\}$ and $\mathbb{H}_l^n = \bigcup_{\mathcal{H} \in \mathbb{H}} \mathcal{H}_l^n$. Define $P(h_l) = \int_{X \times Y} h_l(x, y) \mathrm{d}P(x, y)$ and $P(\mathcal{H}_l) = \inf_{h_l \in \mathcal{H}_l} P(h_l)$ where $P$ is a probability measure on $Z$, and we have $P(h_l) = er_P(h)$ and $er_P(\mathcal{H}) = P(\mathcal{H}_l)$.*

**Definition 2** *Let $(M, d)$ be a pseudo-metric[3] space. $\forall \epsilon > 0$, a subset $\hat{T}$ is called an $\epsilon$-cover of $T \subseteq M$ if $\forall t \in T$, $\exists t' \in \hat{T}$ such that $d(t, t') \leq \epsilon$. Let $\mathbf{P} = P_1 \times \cdots \times P_n$ be the product measure on $Z^n$. For any $(\mathbb{H}_l^n \ni) \mathbf{h}_l, \mathbf{h}_l' : Z^n \to [0,1]$, define the pseudo-metric $d_{\mathbf{P}}(\mathbf{h}_l, \mathbf{h}_l') = \int_{Z^n} |\mathbf{h}_l(\vec{z}) - \mathbf{h}_l'(\vec{z})| d\mathbf{P}(\vec{z})$. Then for any $\epsilon > 0$, the **covering number** $\mathcal{N}(\epsilon, \mathbb{H}_l^n, d_{\mathbf{P}})$ is defined as $\min\{|T| \big| T$ is an $\epsilon$-cover of $\mathbb{H}_l^n$ under the $d_{\mathbf{P}}$ pseudo-metric$\}$, where $|T|$ is the cardinality of $T$.*

We hope to use the empirical loss $\hat{er}_{\mathbf{z}}(\mathcal{H})$ or $\hat{er}_{\mathbf{P}}(\mathcal{H})$ to approximate the expected loss $er_Q(\mathcal{H})$. Instead of the traditional absolute value function $d(x, y) = |x - y|$, we consider the following metric function $(\nu > 0, x, y \geq 0)$ $d_{\nu}[x, y] = \frac{|x-y|}{x+y+\nu}$ to measure the distance between $x$ and $y$, which is first used by (Haussler, 1992). In Section 4, we will bound the deviation $d_{\nu}[\hat{er}_{\mathbf{z}}(\mathcal{H}), er_Q(\mathcal{H})]$ by bounding $d_{\nu}[\hat{er}_{\mathbf{z}}(\mathcal{H}), \hat{er}_{\mathbf{P}}(\mathcal{H})]$ and $d_{\nu}[\hat{er}_{\mathbf{P}}(\mathcal{H}), er_Q(\mathcal{H})]$, respectively.

### 3.2 Almost Π-Related Tasks

In this section, we first propose a new concept of task relatedness, which is called **almost Π-relatedness**. This notion can be considered as the extension of the 'Π-relatedness' notion proposed by (Ben-David & Schuller, 2003). The distinctions between our proposed task relatedness notion and that of Ben-David & Schuller (2003) can be found in Section C.2 of Appendix C.

**Definition 3** *(Almost Π-Related Tasks) Let Π be a set of transformations $\pi : Z \to Z$ and let $P, P_1$ be probability measures on $Z = X \times Y$. We say that $P, P_1$ are almost Π-related probability measures\tasks, if the following conditions are satisfied:*
*(1) $\exists N, N_1 \subseteq Z$ such that $P(N) = P_1(N_1) = 0$, and*
*(2) $\exists \pi \in \Pi$, $\pi$ is a one-to-one mapping from $(Z\backslash N, P)$ onto $(Z\backslash N_1, P_1)$. $\forall A \subseteq Z\backslash N$, $A$ is $P$-measurable if and only if $\pi(A) = \{\pi(x, y) \big| (x, y) \in A\} \subseteq (Z\backslash N_1)$ is $P_1$-measurable, and*
*(3) $\int_{Z\backslash N} \mathbf{1}_A dP = \int_{Z\backslash N_1} \mathbf{1}_{\pi(A)} dP_1$, where $\mathbf{1}_A$ is the indicator function on the set $A$, and*
*(4) the image $\pi(N)$ is a $P_1$-measurable set, and the inverse image $\pi^{-1}(N_1)$ is a $P$-measurable set.*

Note that condition (4) is a weak requirement only to ensure the measurability as well as the integrability of the mapping $\pi$ (or $\pi^{-1}$) on the measure zero set $N$ (or $N_1$). Actually, to validate whether two tasks (say $P$ and $P_1$) are almost Π-related, the most important step is to find the bijective transformation $\pi$ which satisfies the listed conditions (1)-(3) in Definition 3. Then, we can extend the transformation $\pi$ to the measure zero set $N$ by defining $\pi(N)$ as a $P_1$-measurable set (e.g. a $P_1$-measure zero set) and extend $\pi^{-1}$ to $N_1$ in a similar way, such that the extended transformation satisfies the condition (4). The existence of such transformation between two tasks can be theoretically guaranteed by some general topological properties of the given sample space $Z$, which will be discussed in detail in Section 4.4. We next give a closure property assumption of the space $\mathcal{H}_l \in \mathbb{H}_l$ when the transformation set Π is imposed on $\mathcal{H}_l$. Similar to (Ben-David & Schuller, 2003) which assumes the closure property of any hypothesis space under the transformation Π, we also assume that the closure condition is satisfied by any $\mathcal{H}_l \in \mathbb{H}_l$ for deriving better generalization bound.

**Definition 4** *Let Π be the set of transformations on the complete separable metric space $Z = X \times Y$. We say that the function space $\mathcal{H}_l$ is closed under the transformations of Π, if for any $h_l \in \mathcal{H}_l$, any $\pi \in \Pi$, we have the composition function $h_l \circ \pi \in \mathcal{H}_l$.*

We need to give more explanations to the rationality of the closure property of the function space $\mathcal{H}_l$, as the closure property is a very important assumption to derive our generalization bounds. Actually, the almost Π-relatedness defined in Definition 3 can induce an equivalence relationship between two tasks, according to the existence of one bijective function $\pi \in \Pi$. The function $\pi$ can also induce an equivalence relationship between two functions in the space $\mathcal{H}_l$. That means, if a function space $\mathcal{H}_l$ contains a good solution to one task $P$, then $\mathcal{H}_l$ should also contain a good solution to the almost Π-related task $P_1$ of $P$. Since the two tasks $P$ and $P_1$ are equivalent to some extent, the function space $\mathcal{H}_l$ should simultaneously contain the good solutions to these two similar tasks. With such closure property assumption, the complexity of the function space $\mathcal{H}_l$ is related to the degree of relatedness between different tasks from the environment. If the tasks in the environment are all similar (e.g., almost Π-related), then it is sufficient for $\mathcal{H}_l$ to contain one good function as well as its

---

[3]A pseudo-metric is a metric without the condition that $d(x, y) = 0 \Rightarrow x = y$.

$\Pi$-related variants to generalize well to all tasks. Such closure property assumption of the function space is also considered in Ben-David & Schuller (2003). They have demonstrated that as long as the function space is rich enough to contain some function as well as its equivalent class, then the closure property assumption can be fulfilled. Thus in this work we assume such basic closure assumption of function space $\mathcal{H}_l$ holds to derive novel theoretical insights for meta-learning.

We introduce another concept induced from the $\pi$-related tasks, called *almost $\Pi$-related environment*, and its theoretical properties, which will be used to derive our theoretical analysis in Section 4.

**Definition 5** *(Almost $\Pi$-related Environment) In meta learning set up, an environment $(\mathcal{P}, Q)$ on $X \times Y$ is called an **almost $\Pi$-related environment**, if there exists a common probability measure $P$ on $X \times Y$, such that for any measures $P_i \in \mathcal{P}(i \in \mathcal{I}, \mathcal{I}$ is the index set), $P$ and $P_i$ are almost $\Pi$-related in the sense of Definition 3.*

**Lemma 1** *Let $(\mathcal{P}, Q)$ be an almost $\Pi$-related environment on $X \times Y$, $\mathcal{H}_l$ a function space on $X \times Y$. Let $P$ be the underlying common distribution as defined in Definition 5, and $N$ be the $P$-measure zero set as defined in Definition 3. Then for any $P$-measurable set $A \subseteq Z \backslash N$, for any $P_i \in \mathcal{P}(i \in \mathcal{I}), h_l \in \mathcal{H}_l$, we have $P(A) = P_i(\pi_i(A)), P_i(h_l \circ \pi_i^{-1}) = P(h_l)$, where $\pi_i$ is the corresponding transformation between $\Pi$-related tasks $P$ and $P_i$ (as defined in Definition 3).*

**Lemma 2** *Let $(\mathcal{P}, Q)$ be an almost $\Pi$-related environment on $X \times Y$, $P$ be the common distribution that is almost $\Pi$-related to any distribution from $\mathcal{P}$. If a function space $\mathcal{H}_l$ (on $X \times Y$) is closed under the transformations of $\Pi$, then for any probability measure $P_i \in \mathcal{P}$, $er_P(\mathcal{H}) = er_{P_i}(\mathcal{H})$.*

We provide two insights into the theoretical result in Lemma 2 to further explain the motivation of our proposed task relatedness notion: (1) To fully utilize all training samples, two different tasks need to be similar enough so that the hypothesis space which performs well on one of these two tasks can also perform well on another. In other words, the 'best' performance of the hypothesis space need to be similar in both tasks, which is the result in above lemma. (2) To achieve the goal in (1), the hypothesis need to be large enough to contain good solutions that are invariant to the transformation between different tasks. Hence, insight (1) motivates us to define the $\Pi$-relatedness notion to measure the similarity between two tasks in Definition 3, and insight (2) motivates us to assume the closeness property assumption of the hypothesis space in Definition 4. More explanations for the motivation of our proposed task relatedness concept can be found in Appendix A.

## 4 THEORETICAL RESULTS

We first give a novel generalization bound for meta learning in the almost $\Pi$-related environment in Section 4.1. This is particularly achieved by employing the theoretical properties of the almost $\Pi$-related environment in Lemma 2. As shown in Theorem 6 in Section 4.4, assuming the environment to be almost $\Pi$-related is reasonable when the sample space $Z$ possesses the completeness and separability (which are general topological properties satisfied by many metric spaces). Section 4.2 derives a covering number bound for representation learning, which is an active area of meta learning. In Section 4.3, by bounding the covering number with the Lipschitz constants of the matrix and nonlinearity in each layer of the deep neural network, we further establish a spectrally-normalized bound that is independent on the number of the total parameters in neural network in Theorem 5. Moreover, we apply this theoretical result to analyze several practical scenarios like binary\multiclass classification and regression problems under the meta learning framework. The three main technical novelties of our work have been clarified in Remark 2 of Appendix C, to show the technical contributions of this work and outgoing research directions for meta-learning.

### 4.1 A COVERING NUMBER BOUND FOR META LEARNING IN ALMOST $\Pi$-RELATED ENVIRONMENT

To bound $|\hat{er}_{\mathbf{z}}(\mathcal{H}) - er_Q(\mathcal{H})|$, we choose to bound the deviation $d_\nu[\hat{er}_{\mathbf{z}}(\mathcal{H}), \hat{er}_{\mathbf{P}}(\mathcal{H})]$ and $d_\nu[\hat{er}_{\mathbf{P}}(\mathcal{H}), er_Q(\mathcal{H})]$, respectively. We first give an explicit PAC-style generalization bound on $d_\nu[\hat{er}_{\mathbf{z}}(\mathcal{H}), \hat{er}_{\mathbf{P}}(\mathcal{H})]$, which can be considered as an inference of Theorem 18 in (Baxter, 2000).

**Theorem 1** *Let $\mathbb{H}_l^n \subseteq \mathbb{H}_l \oplus \cdots \oplus \mathbb{H}_l$ be the permissible[4] class of functions mapping $(X \times Y)^n$ into [0,1], where $\mathbb{H}_l = \{h_l : h \in \mathcal{H} : \mathcal{H} \in \mathbb{H}\}$, $\oplus$ means direct product. Let $\mathbf{z}$ be generated by $m \geq 2/(\nu \epsilon^2)$ independent trials from $(X \times Y)^n$ according to the product measure $\mathbf{P} = P_1 \times \cdots \times P_n$. For any $\nu > 0, 0 < \epsilon < 1$, for any $\mathcal{H} \in \mathbb{H}$, with probability at least $1 - \delta$ over $\mathbf{z}$, we have*

$$d_\nu[\hat{er}_\mathbf{z}(\mathcal{H}), \hat{er}_\mathbf{P}(\mathcal{H})] \leq \sqrt{\frac{8}{\nu mn} \ln \frac{4\mathcal{N}(\epsilon\nu/8, \mathbb{H}_l^n, d_{\bar{\mathbf{z}}})}{\delta}},$$

*where $\bar{\mathbf{z}} \in Z^{(2m \times n)}$, the top half of $\bar{\mathbf{z}}$ is actually $\mathbf{z}$ itself and the bottom half of $\bar{\mathbf{z}}$ is the copy of $\mathbf{z}$. $\forall \mathbf{h}_l, \mathbf{h}_l' \in \mathbb{H}_l^n, d_{\bar{\mathbf{z}}}(\mathbf{h}_l, \mathbf{h}_l') = 1/2m \sum_{i=1}^{2m} |\mathbf{h}_l(\bar{\mathbf{z}}_i) - \mathbf{h}_l'(\bar{\mathbf{z}}_i)|$.*

On the other hand, we can actually bound the deviation $d_\nu[\hat{er}_\mathbf{P}(\mathcal{H}), er_Q(\mathcal{H})]$ as follow, by using the theoretical properties of the almost $\Pi$-related environment in Lemma 2.

**Theorem 2** *Let $(\mathcal{P}, Q)$ be an almost $\Pi$-related environment on the complete separable metric space $Z = X \times Y$. Let $\mathbf{P} \in \mathcal{P}^n$ be generated by $n$ independent trials from $\mathcal{P}$ according to some product probability measure $Q^n$. Then for any $\nu > 0$, any $\mathcal{H} \in \mathbb{H}$, we have $d_\nu[\hat{er}_\mathbf{P}(\mathcal{H}), er_Q(\mathcal{H})] = 0$.*

Combining Theorem 1 and the Theorem 2, we can obtain a novel generalization bound of convergence rate $\mathcal{O}(\frac{1}{\sqrt{nm}})$ on $|\hat{er}_\mathbf{z}(\mathcal{H}) - er_Q(\mathcal{H})|$ that fully utilizes all $mn$ training data under the proposed meta learning framework in (Baxter, 2000), uniformly over the set of all hypothesis space $\mathcal{H}$.

**Theorem 3** *Let $(\mathcal{P}, Q)$ be an almost $\Pi$-related environment on the complete separable metric space $Z = X \times Y$. Let $\mathbf{z}$ be an $(m, n)$-sample generated by the process described in Section 3.1. Let $\mathbb{H} = \{\mathcal{H}\}$ be any permissible hypothesis space family. Then for any $\mathcal{H} \in \mathbb{H}$, $\epsilon \in (0, 1)$, with probability at least $1 - \delta$ over $\mathbf{z}$, we have*

$$|\hat{er}_\mathbf{z}(\mathcal{H}) - er_Q(\mathcal{H})| \leq \sqrt{\frac{64}{mn} \ln \frac{4\mathcal{N}(\epsilon/4, \mathbb{H}_l^n, d_{\bar{\mathbf{z}}})}{\delta}}.$$

Combining the above theorem and Lemma 2, we subsequently give a corollary which reveals the relationship between task-related meta learning setting and i.i.d. single task learning setting.

**Corollary 1** *In the setting of Theorem 3, let $P_{n+1}$ be the new task sampled from the environment. Then for any $\mathcal{H} \in \mathbb{H}$, $\epsilon \in (0, 1)$, with probability at least $1 - \delta$ over $\mathbf{z}$, we have*

$$er_{P_{n+1}}(\mathcal{H}) \leq \hat{er}_\mathbf{z}(\mathcal{H}) + \sqrt{\frac{64}{mn} \ln \frac{4\mathcal{N}(\epsilon/4, \mathbb{H}_l^n, d_{\bar{\mathbf{z}}})}{\delta}}.$$

There are two important points to note about Corollary 1:
**(1)** If the hypothesis space $\mathcal{H}$ contains only one hypothesis and all probability measures in the environment are the same, then the we can choose the bijective transformation $\pi$ between different distributions as the identity transformation, and the result in Corollary 1 degrades to the traditional covering number based generalization bound in single task learning (see Chapter 10 in (Anthony & Bartlett, 2002)), which utilizes all $mn$ i.i.d. training data. In this sense, meta learning theory can be considered as the extension of the conventional single task learning theory.
**(2)** When bounding the generalization error of a hypothesis space $\mathcal{H}$ under Baxter's proposed meta learning framework, even though different tasks may have different distributions, we can still properly estimate the expected loss on the new task according to the empirical loss on the training tasks, and fully leverage all $mn$ (not necessarily i.i.d.) training data with the use of assumed task relatedness in the environment. Actually, the relatedness condition can be satisfied by tasks that are focused on the complete separable sample space (see more explanations in Section 4.4).

## 4.2 COVERING NUMBER META LEARNING BOUNDS WITH REPRESENTATION LEARNING

In single-task learning setup, representation learning aims to find a good feature embedding $f \in \mathcal{F}$ (e.g., deep neural network) which maps the input space $X$ into the feature space $V$. A prediction

---

[4]Permissibility (Pollard, 1984) is a weak measure-theoretic condition satisfied by almost all "real-world" hypothesis space families. Without loss of generality, we assume that all hypothesis space families discussed through out this paper are permissible .

function $g \in \mathcal{G}$ then projects the feature space $V$ into the output space $Y$. Hence, the hypothesis space can be written as $\mathcal{H} = \mathcal{G} \circ f$, $(f \in \mathcal{F})$. Further, let $\mathbb{H}_l = \{\mathcal{H}_l\}$, each $\mathcal{H}_l = \mathcal{G}_l \circ f$, $f \in \mathcal{F}$, where $\mathcal{G}_l = \{g_l\}$, $g_l = l \circ g$ is the composition function of the prediction function $g$ and the loss function $l$. We will omit the subscript $l$ of $g_l \in \mathcal{G}_l$ for simplicity where the context is clear. In meta learning setup, representation learning chooses to learn a common feature embedding across $n$ training tasks and learn $n$ task-specific functions respectively for $n$ tasks, so we can denote the hypothesis space family $\mathbb{H}_l^n = \{g_1 \circ f \oplus \cdots \oplus g_n \circ f : g_1, ..., g_n \in \mathcal{G}_l, f \in \mathcal{F}\} = \{g_1 \oplus \cdots \oplus g_n \circ \bar{f} : g_1 \oplus \cdots \oplus g_n \in \mathcal{G}_l^n, \bar{f} \in \bar{\mathcal{F}}\} = \mathcal{G}_l^n \circ \bar{\mathcal{F}}$, by defining $\bar{\mathcal{F}} \ni \bar{f} : (X \times Y)^n \to (V \times Y)^n$ with $\bar{f}(x_1, y_1, ..., x_n, y_n) = (f(x_1), y_1, \cdots, f(x_n), y_n)$. We then define two pseudo-metrics over feature embedding space $\bar{\mathcal{F}}$ and prediction function space $\mathcal{G}_l^n$ respectively.

**Definition 6** $\forall \bar{\mathbf{z}} = (z_{ij})_{2m \times n} \in Z^{(2m \times n)}$, define the empirical measure $P_{\bar{\mathbf{z}}}$ on $Z^n$ which puts point mass $1/2m$ on each row $\bar{\mathbf{z}}_i$, $i = 1, ..., 2m$. $\forall \mathbf{s} = ((f(x_{ij}), y_{ij}))_{2m \times n} = ((v_{ij}, y_{ij}))_{2m \times n}$, where $f \in \mathcal{F}$, define an empirical measure $P_{\mathbf{s}}$ which puts mass $1/2m$ on each row $\mathbf{s}_i$, $i = 1, ..., 2m$. $\forall \bar{f}, \bar{f}' \in \bar{\mathcal{F}}$, define the pseudo-metric $d_{[P_{\bar{\mathbf{z}}}, \mathcal{G}_l^n]}(\bar{f}, \bar{f}') = 1/2m \sum_{i=1}^{2m} \sup_{g \in \mathcal{G}_l^n} |g \circ \bar{f}(\bar{\mathbf{z}}_i) - g \circ \bar{f}'(\bar{\mathbf{z}}_i)|$, and $\forall g, g' \in \mathcal{G}_l^n$, define the pseudo-metric $d_{P_{\mathbf{s}}}(g, g') = 1/2m \sum_{i=1}^{2m} |g(\mathbf{s}_i) - g'(\mathbf{s}_i)|$.

Then the following two propositions bound the covering number of the hypothesis space family $\mathbb{H}_l^n$ with two covering numbers that are both defined over the single task.

**Proposition 1** Let $(\bar{\mathcal{P}}, \bar{Q})$ be an environment on $V \times Y$. Denote $\mathcal{N}(\epsilon, \mathcal{G}_l^n, 2m) = \sup_{P \sim \bar{Q}^n} \sup_{\mathbf{s} \sim P^{2m}} \mathcal{N}(\epsilon, \mathcal{G}_l^n, d_{P_{\mathbf{s}}})$. Then for any $\epsilon = \epsilon_1 + \epsilon_2$, any $\bar{\mathbf{z}} \in Z^{2m \times n}$, we have $\mathcal{N}(\epsilon_1 + \epsilon_2, \mathbb{H}_l^n, d_{\bar{\mathbf{z}}}) \leq \mathcal{N}(\epsilon_1, \bar{\mathcal{F}}, d_{[P_{\bar{\mathbf{z}}}, \mathcal{G}_l^n]}) \mathcal{N}(\epsilon_2, \mathcal{G}_l^n, 2m)$.

**Proposition 2** Let $\bar{\mathbf{z}}_{;j}$ be the $j$-th column of the data matrix $\bar{\mathbf{z}}$. Let $\mathcal{N}(\epsilon, \mathcal{G}_l, 2m) = \sup_{P \sim \bar{Q}} \sup_{\bar{s} \sim P^{2m}} \mathcal{N}(\epsilon, \mathcal{G}_l, d_{P_{\bar{s}}})$, where $d_{P_{\bar{s}}}(g, g') = 1/2m \sum_{i=1}^{2m} |g(s_i) - g'(s_i)|$, $\forall g, g' \in \mathcal{G}_l$, $\bar{s} = (s_1, ..., s_{2m})$. Then for any $\epsilon > 0$, we have $\mathcal{N}(\epsilon, \bar{\mathcal{F}}, d_{[P_{\bar{\mathbf{z}}}, \mathcal{G}_l^n]}) \leq \max_{1 \leq j \leq n} \mathcal{N}(\epsilon, \mathcal{F}, d_{[P_{\bar{\mathbf{z}}_{;j}}, \mathcal{G}_l]})$, and $\mathcal{N}(\epsilon, \mathcal{G}_l^n, 2m) \leq \mathcal{N}(\epsilon, \mathcal{G}_l, 2m)^n$.

Recalling Theorem 3 and Propositions 1-2, we can further establish the following covering number bound for meta learning with the representation learning.

**Theorem 4** Let $(\mathcal{P}, Q)$ be an almost $\Pi$-related environment on the complete separable metric space $Z = X \times Y$. Let $\mathbb{H} = \{\mathcal{H}\}$ be the set of hypothesis spaces of the form $\mathcal{H} = \mathcal{G} \circ f$, $f \in \mathcal{F}$. Then for any $\mathcal{H} \in \mathbb{H}$, any $0 < \epsilon < 1$, with probability at least $1 - \delta$ over $\mathbf{z}$, we have

$$|\hat{er}_{\mathbf{z}}(\mathcal{H}) - er_Q(\mathcal{H})| \leq \sqrt{\frac{64}{mn} \ln \frac{4}{\delta}} + \sqrt{\frac{64}{mn} \left( \max_{1 \leq j \leq n} \ln \mathcal{N}(\frac{\epsilon}{8}, \mathcal{F}, d_{[P_{\bar{\mathbf{z}}_{;j}}, \mathcal{G}_l]}) + n \ln \mathcal{N}(\frac{\epsilon}{8}, \mathcal{G}_l, 2m) \right)}$$

We need to highlight the important role of Theorem 4: the covering number of the hypothesis space family $\mathbb{H}_l^n = \mathcal{G}_l^n \circ \bar{\mathcal{F}}$ for meta learning (over $n$ tasks) in Theorem 3 is *converted* into the multiplication of the covering number of the class $\mathcal{F}$ of the feature embeddings (over one task) and the $n$-th power of the covering number of the class $\mathcal{G}_l$ of the prediction functions (over one task). This means that we can introduce recent covering number based theoretical results of deep neural network from single task learning into meta learning (see Remark 2). In particular, by bounding the covering number in Theorem 4 with the Lipschitz constants of the function in each layer of the deep neural network, we can achieve non-parameter-count-based bounds (or say norm-constraint-based bounds (Neyshabur et al., 2017)) for meta learning, which will be detailed in the next section.

### 4.3 SPECTRALLY-NORMALIZED BOUNDS FOR META LEARNING WITH NEURAL NETWORK

Based on the theoretical results in Section 4.2, we now aim to derive non-parameter-count-based spectrally-normalized bounds for meta learning with deep neural network. We consider the $L$-layer depth fully-connected networks with nonlinearities (e.g., activation functions, pooling operators) for each layer, which computes an embedding function $f : \mathbb{R}^{d_0} \to \mathbb{R}^d$, where $d_0, d$ are the dimension of input data and embedded feature, respectively. Each layer of the network has a weight matrix $A_i$ and a $\rho_i$-Lipschitz nonlinearity $\sigma_i$, with $\sigma_i(0) = 0$. Then the composition function is given as

$\sigma_i \circ A_i : \mathbb{R}^{d_{i-1}} \to \mathbb{R}^{d_i}, \forall i \in [L]$. For any $\mathcal{A} = (A_1, \cdots, A_L)$, any input data $x \in X$, define $f_{\mathcal{A}}(x) = \sigma_L(A_L \sigma_{L-1}(A_{L-1} \cdots \sigma_1(A_1 x) \cdots)) \in \mathbb{R}^d$. Let $\mathcal{F}$ denote the class of functions computed by the corresponding networks, and $D$ the maximum of $\{d_0, d_1, ..., d_{L-1}, d\}$.

Further, define a sequence of reference matrix $(M_1, ..., M_L)$ with the same dimensions as $(A_1, ..., A_L)$, where $M_i = 0$ in AlexNet (Krizhevsky et al., 2012) and $M_i = I$ in ResNet (He et al., 2016) to ensure good generalization performances. Let $\|\cdot\|_\sigma$ denote the spectral norm, and let $\|\cdot\|_{p,q}$ denote the $(p, q)$-norm defined by $\|A\|_{p,q} = \big\| (\|A_{:1}\|_p, ..., \|A_{:k}\|_p) \big\|_q$ for matrix $A \in \mathbb{R}^{d \times k}$. We next give a spectrally-normalized meta-learning bound with deep neural network. For the ease of presentation, we bound $\mathcal{O}(\sqrt{\frac{C_1 + nC_2}{nm}})$ with $\mathcal{O}(\sqrt{\frac{C_1}{nm}} + \sqrt{\frac{C_2}{m}})$.

**Theorem 5** *Let $(\mathcal{P}, Q)$ be an almost $\Pi$-related environment on the complete separable metric space $Z = X \times Y$. Let $\mathbb{H} = \{\mathcal{H}_{\mathcal{A}}\} = \{\mathcal{G} \circ f_{\mathcal{A}} : f_{\mathcal{A}} \in \mathcal{F}, \mathcal{A} = (A_1, ..., A_L), \|A_i\|_\sigma \le s_i, \|A_i^\top - M_i^\top\|_{2,1} \le b_i, i \in [L]\}$ be a hypothesis space family where each $\mathcal{H}_{\mathcal{A}}$ is of the form $\mathcal{H}_{\mathcal{A}} = \mathcal{G} \circ f_{\mathcal{A}} = \{g \circ f_{\mathcal{A}}(\cdot) : g = \sigma \circ W, W \in \mathbb{R}^{k \times d}, \|W^\top\|_{2,1} \le \theta\}$, where $\sigma$ is an element-wise function with Lipschitz constant $\theta_\sigma$. Suppose that $\exists b > 0$, for any $x \in X \subseteq \mathbb{R}^{d_0}, \|x\|_2 \le b$. Suppose that the loss function $l$ satisfies two conditions: (1) when composed with $g$, $g_l(\cdot, y)$ is an $\alpha$-Lipschitz function w.r.t. the norm $\|\cdot\|_2$, $\forall$ fixed $y \in Y$; (2) $\forall$ fixed $v \in \mathbb{R}^d$, fixed $y \in Y$, $\forall g = \sigma \circ W, g' = \sigma \circ W' \in \mathcal{G}$, $\exists \beta > 0$, such that $|g_l(v, y) - g_l'(v, y)| \le \beta \|Wv - W'v\|_2$. Then for any $\mathcal{H}_{\mathcal{A}} \in \mathbb{H}$, for any $0 < \epsilon < 1/8$, with probability at least $1 - \delta$ over $\mathbf{z}$, we have*

$$|\hat{er}_{\mathbf{z}}(\mathcal{H}_{\mathcal{A}}) - er_Q(\mathcal{H}_{\mathcal{A}})| \le \sqrt{\frac{64}{mn} \ln \frac{4}{\delta}} + \frac{8b \prod_{l=1}^L s_l \rho_l}{\epsilon \sqrt{mn}} \Big[ \alpha \sqrt{\ln(2D^2)} \Big( \sum_{i=1}^L \big(\frac{b_i}{s_i}\big)^{\frac{2}{3}} \Big)^{\frac{3}{2}} + \beta \theta \sqrt{n \ln(2dk)} \Big].$$

When the loss function $l(\cdot, y)$ is a $\gamma$-Lipschitz function w.r.t. the norm $\|\cdot\|_2$, we can set $\alpha = \gamma \theta_\sigma \theta$, $\beta = \gamma \theta_\sigma$. This is a very useful corollary for a large number of general applications, such as the multiclass classification and regression problems in the following subsections. We also show in Remark 1 of Appendix B that the above bound is informative for two-layer neural network.

### 4.3.1 A Spectrally-Normalized Bound for Binary Classification

We first consider the binary classification problem under the meta learning framework, where $Y = \{0, 1\}$. We choose the classical logistic regression model due to its simplicity and wide applicability in binary classification scenarios. Formally, let $\mathcal{G} = \{\sigma \circ w : w \in \mathbb{R}^d, \|w\|_1 \le \theta\}$ [5] be the class of prediction functions, where $\sigma$ is the sigmoid activation function $\sigma(v) = \frac{1}{1+e^{-v}} (\in [0, 1])$. Let $g_l \circ f(x, y) = g_l(v, y)$ be the loss function on $(x, y)$ where $g_l(v, y) = -y \ln(\sigma(w^\top v)) - (1 - y) \ln(1 - \sigma(w^\top v))$ is the cross-entropy loss. We then have the following claim.

**Claim 1** *In the binary classification problem as described above, $\forall y \in Y$, $g_l(\cdot, y)$ is a $2\theta$-Lipshcitz function w.r.t. $l_2$-norm. Further, fix $v \in \mathbb{R}^d, y \in \{0, 1\}$, then we have $\forall w_1, w_2 \in \mathbb{R}^d$, $|l(w_1^\top v, y) - l(w_2^\top v, y)| \le 2|w_1^\top v - w_2^\top v|$. Thus, we can set $\alpha = 2\theta, \beta = 2$ in Theorem 5 to obtain a spectrally-normalized bound for binary classification with logistic regression model in meta learning.*

### 4.3.2 A Spectrally-Normalized Bound for Multiclass Classification

We further consider the multiclass classification problem under the meta learning framework, where $Y = \{1, ..., k\}(k \ge 3)$. Let $\mathcal{G} = \{W \in \mathbb{R}^{k \times d} : \|W^\top\|_{2,1} \le \theta\}$ be the class of prediction functions, and $\Phi_\rho \circ \mathcal{M}(g \circ f(x), y)(f \in \mathcal{F}, g \in \mathcal{G})$ be the loss on $(x, y)$. $\Phi_\rho(v) = \min(1, \max(0, 1 - \frac{v}{\rho}))$ is called the margin loss and $\rho$ is a positive margin parameter. $\mathcal{M}(g \circ f(x), y) = \max_{j \ne y} g \circ f(x)[j] - g \circ f(x)[y]$ is defined as the margin on $(x, y)$. From Lemma A.3 in (Bartlett et al., 2017) about the Lipschitz property of the margin loss, we have another claim.

**Claim 2** *In the multiclass classification problem as described above, for any $y \in Y$, the loss function $l(\cdot, y) = \Phi_\rho \circ \mathcal{M}(\cdot, y)$ is $2/\rho$-Lipschitz w.r.t. the norm $\|\cdot\|_2$. Then we can set $\gamma = 2/\rho$ and $\theta_\sigma = 1$ ($\sigma$ is the identity map in this case) in Theorem 5 and derive a spectrally-normalized margin bound for multiclass classification in meta learning.*

---

[5] In the binary classification, $\mathcal{H}_{\mathcal{A}} = \mathcal{G} \circ f_{\mathcal{A}}$ can be considered as the set of the functions from $X \to \mathbb{R}$ (not $X \to Y = \{0, 1\}$), but the loss functions $er_Q(\mathcal{H}_{\mathcal{A}})$ and $\hat{er}_{\mathbf{z}}(\mathcal{H}_{\mathcal{A}})$ are still well-defined. We use this notation just for simplicity and concision. Similar treatment can also be found in the multiclass classification problem.

### 4.3.3 A SPECTRALLY-NORMALIZED BOUND FOR REGRESSION

For completeness, we finally consider the regression problem under the meta learning framework, where $Y = [0, 1]$. Let $\mathcal{G} = \{\sigma \circ w : w \in \mathbb{R}^d, \|w\|_1 \le \theta\}$ be the class of prediction functions, where $\sigma$ is the sigmoid activation function $\sigma(v) = \frac{1}{1+e^{-v}} (\in [0, 1])$ with the Lipschitz constant $1/4$ (note that the derivative $\sigma'(v) = \sigma(v)(1 - \sigma(v)) \le 1/4$). Let the loss function $l$ be the squared loss function by defining $g_l \circ f(x, y) = (\sigma(w^\top f(x)) - y)^2 (f \in \mathcal{F}, g \in \mathcal{G})$.

**Claim 3** *In the regression problem as described above, for any $y \in Y$, the loss function $l(\cdot, y)$ is 2-Lipschitz w.r.t. $\|\cdot\|_2$. Then we can set $\gamma = 2$ and $\theta_\sigma = 1/4$ ($\sigma$ is the sigmoid function in this case) in Theorem 5 and derive a spectrally-normalized bound for regression problem in meta learning.*

### 4.4 WHEN THE ENVIRONMENT IS ALMOST $\Pi$-RELATED?

In this section, we explore when the given environment $(\mathcal{P}, \mathcal{Q})$ is almost $\Pi$-related, i.e., whether there exists a common underlying distribution $P$ that is almost $\Pi$-related with any measure $P_i$ sampled from the environment. As the discussion below the Definition 3 shows, the most crucial step is to find an (almost) bijective transformation $\pi$ which satisfies the conditions (1)-(3) in Definition 3. We claim that this bijective transformation is actually equivalent to the *almost isomorphism* between two probability measure spaces (Chapter 9 in (Bogachev, 2007)). We first give the definition of the almost isomorphism and the proof of Theorem 6 can be found in the Appendix.

**Definition 7** *(Almost Isomorphism) Let $(Z_1, \mathcal{A}, \mu)$ and $(Z_2, \mathcal{B}, \nu)$ be two measure spaces.*
*(1) A* point isomorphism *$\pi$ of these spaces is a one-to-one mapping of $Z_1$ on to $Z_2$ such that $\mu \circ \pi^{-1} = \nu$ and $\pi(\mathcal{A}) = \mathcal{B}$. That is, $\forall A \in \mathcal{A}, \pi(A) \in \mathcal{B}$, and vice versa.*
*(2) $(Z_1, \mathcal{A}, \mu)$ and $(Z_2, \mathcal{B}, \nu)$ are called* almost isomorphic *if there exist sets $N_1 \in \mathcal{A}_\mu, N_2 \in \mathcal{B}_\nu$ with $\mu(N_1) = \nu(N_2) = 0$ and a point isomorphism $\pi$ of the spaces $Z_1 \backslash N_1$ and $Z_2 \backslash N_2$ that are equipped with the restriction of the measures $\mu$ and $\nu$ and the complete $\sigma$-algebra $\mathcal{A}_\mu$ and $\mathcal{B}_\nu$.*

With elaborate treatment, we can reveal the existence of the almost isomorphism between any two complete separable metric spaces. That means, there exists a common distribution $P$ that is almost $\Pi$-related to any distribution sampled from the same environment. It is formally stated as below.

**Theorem 6** *Let $(\mathcal{P}, \mathcal{Q})$ be an environment on the complete separable metric space $Z$. Then for any atomless $P_i, P_j \in \mathcal{P}(i \ne j, i, j \in \mathcal{I})$, the probability measure spaces $(Z, \mathcal{B}_i, P_i)$ and $(Z, \mathcal{B}_j, P_j)$ are almost isomorphic. In other words, the two measures $P_i$ and $P_j$ are almost $\Pi$-related.*

From Theorem 6 we have seen that, any two atomless probability measures $P_i, P_j (i \ne j)$ on the complete separable metric space $Z$ are almost $\Pi$-related. Therefore, we can choose $P_1 \in \mathcal{P}$ as the common distribution that is almost $\Pi$-related to any probability measure $P_i (i \in \mathcal{I})$ sampled from the set $\mathcal{P}$. Hence, we demonstrate the existence of the common distribution $P$ defined in Definition 5. Further, we can drop the condition of the $\Pi$-relatedness of all tasks sampled from the environment $(\mathcal{P}, \mathcal{Q})$ in Theorems 2-5, since this task relatedness requirement is fulfilled by the fact that $Z$ is a complete separable metric space. We also need to point out that, the completeness and separability are both very general topological properties that can be satisfied by a number of metric spaces such as the real space $\mathbb{R}^d$, the closed subspace in $\mathbb{R}^d$, and the product space of the complete separable metric spaces. For example, in $d$-dimensional regression problem, the sample space $Z = \mathbb{R}^d \times [a, b]$ $(a, b \in \mathbb{R})$ is also a complete separable metric space.

## 5 CONCLUSIONS

This paper provides a covering number based generalization bound for meta learning by exploiting the task relatedness of the environment. When analyzing the meta learning with deep neural network, we derive spectrally-normalized bounds for classification and regression problems. Our bounds rely on two basic assumptions: the relatedness between different tasks and the closure property of the hypothesis space. We demonstrate that, the first task-relatedness assumption can be satisfied if the sample space is a complete separable metric space. We also show that the closure property assumption holds when the hypothesis space contains good solutions as well as their equivalent variants. Our ongoing research includes analyzing the convolutional neural network as well as establishing sharper generalization bounds for meta learning via algorithmic analysis.

ACKNOWLEDGEMENTS

Jiechao Guan sincerely thanks Dr. Qi Meng from MSRA for helpful discussions and insightful comments on the writing of this paper. We thank anonymous reviewers for spotting a mistake in the original proof of this paper. We also thank all reviewers for their constructive suggestions to improve the quality of this paper. This work was supported in part by National Natural Science Foundation of China (61976220 and 61832017), Beijing Outstanding Young Scientist Program (BJJWZYJH012019100020098), China Unicom Innovation Ecological Cooperation Plan, and Large-Scale Pre-Training Program 468 of Beijing Academy of Artificial Intelligence (BAAI).

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

# APPENDIX

## A    MOTIVATION OF TASK RELATEDNESS

We simply explain our motivation of exploring 'task relatedness' by comparing the differences between single task learning and meta learning in Table 1, to fully utilize all training samples.

Table 1: The differences between traditional single task learning and meta learning framework proposed by Baxter (Baxter, 2000), where $er_Q(\mathcal{H}) - er_{\mathbf{z}}(\mathcal{H}) = \int \inf_{h \in \mathcal{H}} er_P(h) \mathrm{d}Q(P) - \sum_{i=1}^{n} \inf_{h \in \mathcal{H}} \hat{er}_{P_i}(h)$. The proposal of our task relatedness notion comes from the motivation that probabilities measures $\{P_i\}_{i=1}^{n}$ sampled from the environment $(\mathcal{P}, \mathcal{Q})$ need to be **equivalent**, namely, **almost $\Pi$-related**.

| Setting | Single-Task Learning | Meta-Learning |
|---|---|---|
| **Goal** | choose hypothesis $h \in \mathcal{H}$ | choose hypothesis space $\mathcal{H} \in \mathbb{H}$ |
| **Dealing Objective** | i.i.d. sample $\vec{z} = \{z_i\}_{i=1}^{m}$ | i.i.d. probability measures $\{P_i\}_{i=1}^{n}$ |
| **Generalization Gap** | $er_P(h) - \hat{er}_{\vec{z}}(h)$ | $er_Q(\mathcal{H}) - er_{\mathbf{z}}(\mathcal{H})$ |
| **Requirement** | $\{z_i\}_{i=1}^{m}$ are similar enough i.e., identical distributed samples. | $\{P_i\}_{i=1}^{n}$ need to be similar enough i.e., **equivalent measures**. |

## B    DETAILED PROOFS OF OUR THEORETICAL RESULTS

### B.1    PROOF OF THEORETICAL PROPERTIES OF ALMOST $\Pi$-RELATED ENVIRONEMENT

***Proof of Lemma 1 in the main paper.***
From Definition 3, $\exists N_i \subseteq Z$, such that $P_i(N_i) = 0$. Then for any $P$-measurable set $A \subseteq Z \backslash N$,

$$P(A) = \int_Z \mathbf{1}_{A \cap (Z \backslash N)} + \mathbf{1}_{A \cap N} \mathrm{d}P = \int_{Z \backslash N} \mathbf{1}_A \mathrm{d}P$$

$$= \int_{Z \backslash N_i} \mathbf{1}_{\pi_i(A)} \mathrm{d}P_i = \int_Z \mathbf{1}_{\pi_i(A)} \mathrm{d}P_i = \int_Z \mathbf{1}_{A \circ \pi_i^{-1}} \mathrm{d}P_i = \int_Z \mathbf{1}_A \mathrm{d}P_i \circ \pi_i = P_i(\pi_i(A)),$$

where the third equality holds due to the condition (3) in Definition 3, and the sixth equality holds due to the equivalent integral transformation $\int f \circ g^{-1} \mathrm{d}P = \int f \mathrm{d}P \circ g$. Similarly, recalling $P(h_l)$ from Definition 1 and noticing $\pi_i^{-1}(N_i)$ is $P$-measurable from condition (4) in Definition 3, we have

$$P_i(h_l \circ \pi_i^{-1}) = (\int_{Z \backslash N_i} + \int_{N_i}) h_l \circ \pi_i^{-1} \mathrm{d}P_i = \int_{Z \backslash N} h_l \mathrm{d}P = P(h_l).$$

The above second equality holds due to the fact that $\int_{N_i} h_l \circ \pi_i^{-1} \mathrm{d}P_i = 0$ (since $P_i(N_i) = 0$ and $h_l$ is a bounded function), and the condition (3) of $\pi_i^{-1}$ in Definition 3. ∎

***Proof of Lemma 2 in the main paper.***
To prove $\inf_{h \in \mathcal{H}} er_P(h) = \inf_{h \in \mathcal{H}} er_{P_i}(h)$, it is equivalent to prove $\inf_{h_l \in \mathcal{H}_l} P(h_l) = \inf_{h_l \in \mathcal{H}_l} P_i(h_l)$, then it suffices to show that $\forall h_l \in \mathcal{H}_l, \exists h_l' \in \mathcal{H}_l$ we have $P_i(h_l') \leq P(h_l)$. Since the symmetricity of $P$ and $P_i$ (i.e., $P \circ \pi_i^{-1} = P_i, P_i \circ \pi_i = P$ holds almost everywhere), we can find another $h_l''$ such that $P(h_l'') \leq P_i(h_l)$. In fact, let $h_l' = h_l \circ \pi_i^{-1}$, from Lemma 1, we have $P_i(h_l') \leq P(h_l)$. ∎

### B.2    PROOF OF THE COVERING NUMBER BOUND FOR META LEARNING IN ALMOST $\Pi$-RELATED ENVIRONMENT

**Lemma 3** *(Theorem 18 in (Baxter, 2000)) Let $\mathcal{H} \subseteq \mathcal{H}_1 \oplus \cdots \oplus \mathcal{H}_n$ be a permissible class of functions mapping $(X \times Y)^n$ into $[0, 1]$. Let $\mathbf{z} \in (X \times Y)^{(m,n)}$ be generated by $m \geq 2/(\epsilon^2 \nu)$ independent*

*trials from $(X \times Y)^n$ according to some product probability measure $\mathbf{P} = P_1 \times \cdots \times P_n$. For any $\nu > 0, 0 < \epsilon < 1$, for any $\mathbf{h} \in \mathcal{H}$, we have*

$$Pr\{\mathbf{z} \in (X \times Y)^{(m,n)} : \sup_{\mathcal{H}} d_\nu[\hat{er}_\mathbf{z}(\mathbf{h}), er_\mathbf{P}(\mathbf{h})] > \epsilon\} \leq 4\mathcal{N}(\epsilon\nu/8, \mathcal{H}, d_{\bar{\mathbf{z}}}) \exp(-\epsilon^2 \nu mn/8).$$

***Proof of Theorem 1 in the main paper.***

$$Pr\{\mathbf{z} \in (X \times Y)^{(m,n)} : \sup_{\mathbb{H}} d_\nu[\hat{er}_\mathbf{z}(\mathcal{H}), \hat{er}_\mathbf{P}(\mathcal{H})] > \epsilon\}$$

$$\leq Pr\{\mathbf{z} \in (X \times Y)^{(m,n)} : \sup_{\mathbb{H}_l^n} d_\nu[\hat{er}_\mathbf{z}(\mathbf{h}), er_\mathbf{P}(\mathbf{h})] > \epsilon\}$$

$$\leq 4\mathcal{N}(\epsilon\nu/8, \mathbb{H}_l^n, d_{\bar{\mathbf{z}}}) \exp(-\epsilon^2 \nu mn/8),$$

where the first inequality holds due to the Lemma 24 in (Baxter, 2000), and the second inequality holds due to the Lemma 3 in the supplementary material. Let the r.h.s. of the above inequality be less than the confidence parameter $\delta \in (0,1)$, then we have $\epsilon^2 \geq \frac{8}{\nu mn} \ln \frac{4\mathcal{N}(\epsilon\nu/8, \mathbb{H}_l^n, d_{\bar{\mathbf{z}}})}{\delta}$, which gives the explicit PAC-style generalization bound on $d_\nu[\hat{er}_\mathbf{z}(\mathcal{H}), \hat{er}_\mathbf{P}(\mathcal{H})]$ in Theorem 1. ∎

***Proof of Theorem 2 in the main paper.***
According to Lemma 2, there exists a common distribution $P \in \mathcal{P}$ such that $er_P(\mathcal{H}) = er_{P_i}(\mathcal{H})$ for any $P_i \in \mathcal{P}$. Then $er_Q(\mathcal{H}) = \int_\mathcal{P} er_{P_i}(\mathcal{H}) dQ(P_i) = \int_\mathcal{P} er_P(\mathcal{H}) dQ(P_i) = er_P(\mathcal{H}) = 1/n \sum_{j=1}^n er_{P_j}(\mathcal{H}) = \hat{er}_\mathbf{P}(\mathcal{H})$, which indicates $|er_Q(\mathcal{H}) - \hat{er}_\mathbf{P}(\mathcal{H})| = 0$. ∎

***Proof of Theorem 3 in the main paper.***
By the fact that the loss function $l$ has range $[0,1]$, and the triangle inequality of metric $d_\nu$,

$$\frac{|\hat{er}_\mathbf{z}(\mathcal{H}) - er_Q(\mathcal{H})|}{\nu + 2} \leq d_\nu[\hat{er}_\mathbf{z}(\mathcal{H}), er_Q(\mathcal{H})]$$

$$\leq d_\nu[\hat{er}_\mathbf{z}(\mathcal{H}), \hat{er}_\mathbf{P}(\mathcal{H})] + d_\nu[\hat{er}_\mathbf{P}(\mathcal{H}), er_Q(\mathcal{H})] \leq \sqrt{\frac{8}{\nu mn} \ln \frac{4\mathcal{N}(\epsilon\nu/8, \mathbb{H}_l^n, d_{\bar{\mathbf{z}}})}{\delta}}.$$

Letting $\nu = 2$ (to minimize $\sqrt{\nu} + \frac{2}{\sqrt{\nu}}$) gives the result. ∎

### B.3 PROOF OF COVERING NUMBER BOUNDS FOR META LEARNING WITH REPRESENTATION LEARNING

***Proof of Proposition 1 in the main paper.***
Fix an empirical measure $P_{\bar{\mathbf{z}}}$ on $(X \times Y)^n$, and let $\widehat{\mathcal{F}}$ be a minimum size $\epsilon_1$-cover for $(\bar{\mathcal{F}}, d_{[P_{\bar{\mathbf{z}}}, \mathcal{G}_l^n]})$, then we have $|\widehat{\mathcal{F}}| = \mathcal{N}(\epsilon_1, \bar{\mathcal{F}}, d_{[P_{\bar{\mathbf{z}}}, \mathcal{G}_l^n]})$, and $\forall \bar{f} \in \bar{\mathcal{F}}, \exists \hat{f} \in \widehat{\mathcal{F}}$ such that $d_{[P_{\bar{\mathbf{z}}}, \mathcal{G}_l^n]}(\bar{f}, \hat{f}) \leq \epsilon_1$. For any $\hat{f} \in \widehat{\mathcal{F}}$, let $P_{\bar{\mathbf{z}}} \circ \hat{f}^{-1}$ be the induced probability measure on $(V \times Y)^n$ by defining $P_{\bar{\mathbf{z}}} \circ \hat{f}^{-1}(A) = P_{\bar{\mathbf{z}}}(\hat{f}^{-1}(A)), \forall A \in \sigma$-algebra on $(V \times Y)^n$. Let $G_{\hat{f}}$ be the minimum size $\epsilon_2$-cover for $(\mathcal{G}_l^n, d_{P_{\bar{\mathbf{z}}} \circ \hat{f}^{-1}})$. Hence $|\mathcal{G}_{\hat{f}}| = \mathcal{N}(\epsilon_2, \mathcal{G}_l^n, d_{P_{\bar{\mathbf{z}}} \circ \hat{f}^{-1}}) \leq \mathcal{N}(\epsilon_2, \mathcal{G}_l^n, 2m)$. Let $N = \{g \circ f : f \in \widehat{\mathcal{F}}$ and $g \in \mathcal{G}_f\}$, then we have $|N| \leq \mathcal{N}(\epsilon_1, \bar{\mathcal{F}}, d_{[P_{\bar{\mathbf{z}}}, \mathcal{G}_l^n]}) \mathcal{N}(\epsilon_2, \mathcal{G}_l^n, 2m)$, which satisfies the required cardinality condition.
It remains to show that $N$ is an $\epsilon_1 + \epsilon_2$-cover of $\mathbb{H}_l^n$. Actually, $\forall g_1 \oplus \cdots \oplus g_n \circ \bar{f} \in \mathbb{H}_l^n$, choose $\hat{f} \in \widehat{\mathcal{F}}$ such that $d_{[P_{\bar{\mathbf{z}}}, \mathcal{G}_l^n]}(\bar{f}, \hat{f}) \leq \epsilon_1$, and choose $g_1' \oplus \cdots \oplus g_n' \in \mathcal{G}_{\hat{f}}$ such that $d_{P_{\bar{\mathbf{z}}} \circ \hat{f}^{-1}}(g_1 \oplus \cdots \oplus g_n, g_1' \oplus \cdots \oplus g_n') \leq \epsilon_2$. Then the triangle inequality of the distance metric $d_{\bar{\mathbf{z}}}$ implies that

$$d_{\bar{\mathbf{z}}}(g_1 \oplus \cdots \oplus g_n \circ \bar{f}, g_1' \oplus \cdots \oplus g_n' \circ \hat{f})$$

$$\leq d_{\bar{\mathbf{z}}}(g_1 \oplus \cdots \oplus g_n \circ \bar{f}, g_1 \oplus \cdots \oplus g_n \circ \hat{f}) + d_{\bar{\mathbf{z}}}(g_1 \oplus \cdots \oplus g_n \circ \hat{f}, g_1' \oplus \cdots \oplus g_n' \circ \hat{f})$$

$$\leq d_{[P_{\bar{\mathbf{z}}}, \mathcal{G}_l^n]}(\bar{f}, \hat{f}) + d_{P_{\bar{\mathbf{z}}} \circ \hat{f}^{-1}}(g_1 \oplus \cdots \oplus g_n, g_1' \oplus \cdots \oplus g_n')$$

$$\leq \epsilon_1 + \epsilon_2.$$

The second inequality holds due to the definition of $d_{[P_{\bar{\mathbf{z}}}, \mathcal{G}_l^n]}(\bar{f}, \hat{f}) = 1/2m \sum_{i=1}^{2m} \sup_{g \in \mathcal{G}_l^n} |g \circ \bar{f}(\bar{\mathbf{z}}_i) - g \circ \hat{f}(\bar{\mathbf{z}}_i)|$ and the equivalent integral transformation $\int g \mathrm{d} P_{\bar{\mathbf{z}}} \circ \hat{f}^{-1} = \int g \circ \hat{f} \mathrm{d} P_{\bar{\mathbf{z}}}, \forall g \in \mathcal{G}_l^n$. ∎

***Proof of Proposition 2 in the main paper.***
For any $\bar{f}, \bar{f}' \in \bar{\mathcal{F}}$, we have

$$
\begin{aligned}
d_{[P_{\bar{\mathbf{z}}}, \mathcal{G}_l^n]}(\bar{f}, \bar{f}') &= \frac{1}{2m} \sum_{i=1}^{2m} \sup_{g \in \mathcal{G}_l^n} |g \circ \bar{f}(\bar{\mathbf{z}}_i) - g \circ \bar{f}'(\bar{\mathbf{z}}_i)| \\
&= \frac{1}{2m} \sum_{i=1}^{2m} \sup_{\mathcal{G}_l^n} |\frac{1}{n} \sum_{j=1}^{n} g_j \circ f(z_{ij}) - g_j \circ f'(z_{ij})| \\
&\leq \frac{1}{n} \sum_{j=1}^{n} \frac{1}{2m} \sum_{i=1}^{2m} \sup_{g_j \in \mathcal{G}_l} |g_j \circ f(z_{ij}) - g_j \circ f'(z_{ij})| \\
&\leq \max_{1 \leq j \leq n} d_{[P_{\bar{\mathbf{z}}_{;j}}, \mathcal{G}_l]}(f, f'),
\end{aligned}
$$

which completes the proof of the first inequality. Similarly, for the second one, $\forall P \sim \bar{Q}^n, \mathbf{s} \sim P^{2m}, \forall g, g' \in \mathcal{G}_l^n$,

$$
\begin{aligned}
d_{P_{\mathbf{s}}}(g, g') &= \frac{1}{2m} \sum_{i=1}^{2m} |g(\mathbf{s}_i) - g'(\mathbf{s}_i)| \\
&= \frac{1}{2m} \sum_{i=1}^{2m} |\frac{1}{n} \sum_{j=1}^{n} g_j(v_{ij}, y_{ij}) - g_j'(v_{ij}, y_{ij})| \\
&\leq \frac{1}{n} \sum_{j=1}^{n} \frac{1}{2m} \sum_{i=1}^{2m} |g_j(v_{ij}, y_{ij}) - g_j'(v_{ij}, y_{ij})| \\
&\leq \max_{1 \leq j \leq n} \frac{1}{2m} \sum_{i=1}^{2m} |g_j(v_{ij}, y_{ij}) - g_j'(v_{ij}, y_{ij})| \\
&= \max_{1 \leq j \leq n} d_{P_{\mathbf{s}_{;j}}}(g_j, g_j').
\end{aligned}
$$

Therefore, $\mathcal{N}(\epsilon, \mathcal{G}_l^n, d_{P_{\mathbf{s}}}) \leq (\max_{1 \leq j \leq n} \mathcal{N}(\epsilon, \mathcal{G}_l, d_{P_{\mathbf{s}_{;j}}}))^n$ and thus $\mathcal{N}(\epsilon, \mathcal{G}_l^n, 2m) \leq \mathcal{N}(\epsilon, \mathcal{G}_l, 2m)^n$. ∎

## B.4 Proof of Spectrally-Normalized Bounds for Meta Learning with Deep Neural Network

To demonstrate Theorem 5 in the main paper, we need to give the following two important lemmas, which gives the non-parameter-count-based spectrally-normalized bounds for deep neural network in the single task learning.

**Lemma 4** *(Bartlett et al., 2017) Let positive reals $(\alpha, \beta, \epsilon)$ and positive integer $k$ be given. Let matrix $\mathbf{V}^\top \in \mathbb{R}^{2m \times d}$ be given with $\|\mathbf{V}^\top\|_2 \leq \beta$. Then*

$$
\mathcal{N}(\{\mathbf{V}^\top A : A \in \mathbb{R}^{d \times k}, \|A\|_{2,1} \leq \alpha\}, \epsilon, \|\cdot\|_2) \leq (2dk)^{\lceil \frac{\alpha^2 \beta^2}{\epsilon^2} \rceil}.
$$

**Lemma 5** *(Bartlett et al., 2017) Let fixed nonlinearities $(\sigma_1, ..., \sigma_L)$ and reference matrices $(M_1, ..., M_L)$ be given, where $\sigma_i$ is $\rho_i$-Lipschitz and $\sigma_i(0) = 0$. Let spectral norm bounds $(s_1, ..., s_L)$, and matrix $(2, 1)$-norm bounds $(b_1, ..., b_L)$ be given. Let the input data matrix $\mathbf{X} \in \mathbb{R}^{2m \times d_0}$ be given, where the $m$ rows correspond to data points. Let $\mathcal{H}_{\mathbf{X}}$ denote the family of matrices obtained by evaluating $\mathbf{X}$ with all choices of network $f_A$ : $\mathcal{H}_{\mathbf{X}} = \{f_A(\mathbf{X}^\top) | \mathcal{A} = $*

$(A_1, ..., A_L), \|A_i\|_\sigma \le s_i, \|A_i^\top - M_i^\top\|_{2,1} \le b_i\}$, *where each matrix has dimension at most $D$ along each axis. Then for any $\epsilon > 0$,*

$$\ln\mathcal{N}(\mathcal{H}_{\mathbf{X}}, \epsilon, \|\cdot\|_2) \le \frac{\|\mathbf{X}\|_2^2 \ln(2D^2)}{\epsilon^2} \Big(\prod_{j=1}^L s_j^2 \rho_j^2\Big) \Big(\sum_{i=1}^L (\frac{b_i}{s_i})^{\frac{2}{3}}\Big)^3$$

***Proof of Theorem 5 in the main paper.***
First, we bound $\ln\mathcal{N}(\epsilon, \mathcal{F}, d_{[P_{\vec{z}}, \mathcal{G}_l]})$. For any $P \in \mathcal{P}, \vec{z} \sim P^{2m}$, for any $f_\mathcal{A}, f_{\mathcal{A}'} \in \mathcal{F}$,

$$d_{[P_{\vec{z}}, \mathcal{G}_l]}(f_\mathcal{A}, f_{\mathcal{A}'}) = \frac{1}{2m} \sum_{i=1}^{2m} \sup_{g_l \in \mathcal{G}_l} |g_l \circ f_\mathcal{A}(z_i) - g_l \circ f_{\mathcal{A}'}(z_i)|$$

$$= \frac{1}{2m} \sum_{i=1}^{2m} \sup_{g_l \in \mathcal{G}_l} |g_l(f_\mathcal{A}(x_i), y_i) - g_l(f_{\mathcal{A}'}(x_i), y_i)|$$

$$\le \frac{\alpha}{2m} \sum_{i=1}^{2m} \|f_\mathcal{A}(x_i) - f_{\mathcal{A}'}(x_i)\|_2 \quad \text{(Lipschitz)}$$

$$\le \frac{\alpha}{\sqrt{2m}} \|f_\mathcal{A}(\mathbf{X}^\top) - f_{\mathcal{A}'}(\mathbf{X}^\top)\|_2. \quad \text{(Jensen)}$$

Applying Lemma 5, we then have

$$\ln\mathcal{N}(\epsilon, \mathcal{F}, d_{[P_{\vec{z}}, \mathcal{G}_l]}) \le \ln\mathcal{N}(\frac{\sqrt{2m}\epsilon}{\alpha}, \mathcal{F}, \|\cdot\|_2)$$

$$\le \frac{\alpha^2 \|\mathbf{X}\|_2^2 \ln(2D^2)}{2m\epsilon^2} \Big(\prod_{j=1}^L s_j^2 \rho_j^2\Big) \Big(\sum_{i=1}^L (\frac{b_i}{s_i})^{\frac{2}{3}}\Big)^3 \le \frac{\alpha^2 b^2 \ln(2D^2)}{\epsilon^2} \Big(\prod_{j=1}^L s_j^2 \rho_j^2\Big) \Big(\sum_{i=1}^L (\frac{b_i}{s_i})^{\frac{2}{3}}\Big)^3. \tag{1}$$

Next, we bound $\ln\mathcal{N}(\epsilon, \mathcal{G}_l, d_{P_{\vec{s}}})$. Actually, for any $P \sim \bar{Q}, \vec{s} \sim P^{2m}$, for any $g_l, g_l' \in \mathcal{G}_l$, using condition (2) of the loss function $l$ and Jensen inequality, we have

$$d_{P_{\vec{s}}}(g, g') = \frac{1}{2m} \sum_{i=1}^{2m} |g_l(v_i, y_i) - g_l'(v_i, y_i)|$$

$$\le \frac{\beta}{2m} \sum_{i=1}^{2m} \|W v_i - W' v_i\|_2 \le \frac{\beta}{\sqrt{2m}} \|\mathbf{V}^\top W^\top - \mathbf{V}^\top W'^\top\|_2,$$

where $\mathbf{V} = (v_1, ..., v_{2m}) \in \mathbb{R}^{d \times 2m}$. Then combining the above results and Lemma 4, we have $\ln\mathcal{N}(\epsilon, \mathcal{G}_l, d_{P_{\vec{s}}})$

$$\le \ln\mathcal{N}(\frac{\sqrt{2m}\epsilon}{\beta}, \{\mathbf{V}^\top W^\top : \|W^\top\|_{2,1} \le \theta\}, \|\cdot\|_2)$$

$$\le \frac{\beta^2 \theta^2 \|\mathbf{V}^\top\|_2^2}{2m\epsilon^2} \ln(2dk) \le \frac{\beta^2 \theta^2 b^2 \prod_{l=1}^L s_l^2 \rho_l^2}{\epsilon^2} \ln(2dk). \tag{2}$$

The last inequality holds due to the Lipschitz property of the activation $\sigma_l$ and the matrix $A_l (l = 1, ..., L), \forall i \in [2m]$

$$\|v_i\|_2 = \|\sigma_L(A_L \cdots \sigma_1 A_1(x_i) \cdots) - \sigma_L(0)\|_2$$

$$\le s_L \rho_L \|\sigma_{L-1}(A_{L-1} \cdots \sigma_1 A_1(x_i) \cdots)\|_2 \le b \prod_{l=1}^L s_l \rho_l.$$

Recalling Theorem 4 in the main paper and Eqs.(1)-(2) in the supplementary gives the result. ∎

**Remark 1** *(A situation where our spectrally-normalized bound for meta-learning is informative)*
We further provide a situation where the neural network is a two-layer network (i.e. composed of a

hidden layer with $D$ units and an output layer) for $k$-class classification problem, to provide more information of our spectrally-normalized bound from two aspects: **(i)** Our bound in Theorem 5 is more informative than the traditional VC-dimension bound. Actually, our bound in Theorem 5 for two-layer network can be rewritten as follow:

$$\sqrt{\frac{64}{mn}\ln\frac{4}{\delta}} + \frac{8bs_1\rho_1}{\epsilon\sqrt{mn}}\Big[\alpha\sqrt{\ln{(2D^2)}}(\frac{b_1}{s_1}) + \beta\theta\sqrt{n\ln{(2dk)}}\Big],$$

which is of order $\mathcal{O}(\frac{b_1\sqrt{\ln D^2}}{\sqrt{nm}} + \frac{s_1\sqrt{\ln{(Dk)}}}{\sqrt{m}})$. Under the same setting, the VC-dimension based meta-learning bound for neural networks (i.e. obtained with techniques from Theorem 8 in Baxter (2000)) is of order $\mathcal{O}(\sqrt{\frac{v}{nm}} + \sqrt{\frac{v}{m}})$. Further note that the VC-dimension of the neural networks is $v \approx W\ln W$ (where $W$ is the total number of the parameters, see Bartlett et al. (2019)), thus the VC-dimension for two-layer neural networks is about $v \approx (D^2 + Dk)\ln{(D^2 + Dk)}$ and the meta-learning bound is about $\mathcal{O}(\sqrt{\frac{(D^2+Dk)\ln{(D^2+Dk)}}{nm}} + \sqrt{\frac{(D^2+Dk)\ln{(D^2+Dk)}}{m}})$, which is looser than our spectrally-normalized meta-learning bound of $\mathcal{O}(\frac{b_1\sqrt{\ln D^2}}{\sqrt{nm}} + \frac{s_1\sqrt{\ln{(Dk)}}}{\sqrt{m}})$. **(ii)** Our bound in Theorem 5 is informative under the implicit regularization framework of SGD. Note that SGD can always find a minimum trace/nuclear norm for neural networks (especially for two-layer networks, see Gunasekar et al. (2017)), therefore the norm parameters $s_1$ and $b_1$ in our bound for two-layer neural networks can be small, and hence our spectrally-normalized meta-learning bound of order $\mathcal{O}(\frac{b_1\sqrt{\ln D^2}}{\sqrt{nm}} + \frac{s_1\sqrt{\ln{(Dk)}}}{\sqrt{m}})$ can be informative.

***Proof of the Corollary under Theorem 5 in the main paper.***
$\forall v_1, v_2 \in \mathbb{R}^d, \forall y \in Y$,

$$|g_l(v_1, y) - g_l(v_2, y)| = |l(g(v_1), y) - l(g(v_2), y)|$$
$$= |l(\sigma \circ Wv_1, y) - l(\sigma \circ Wv_2, y)| \le \gamma\theta_\sigma\|Wv_1 - Wv_2\|_2 \le \gamma\theta_\sigma\theta\|v_1 - v_2\|_2,$$

where the second inequality holds since $\|W^\top\|_{2,1}$ is a kind of matrix norm of $W$. Hence we can set $\alpha = \gamma\theta_\sigma\theta$. Similarly, $\forall g_l, g_l' \in \mathcal{G}_l, v \in \mathbb{R}^d, y \in Y$, we have $|g_l(v, y) - g_l'(v, y)|$

$$= |l(\sigma \circ Wv, y) - l(\sigma \circ W'v, y)| \le \gamma\theta_\sigma\|Wv - W'v\|_2.$$

Then we can obtain $\beta = \gamma\theta_\sigma$. Combining the above results with Theorem 5 gives the result. $\blacksquare$

***Proof of Claim 1 in the main paper.***
$\forall v \in \mathbb{R}^d, g_l(v, y) = -y\ln(\sigma(w^\top v)) - (1 - y)\ln(1 - \sigma(w^\top v))$. Therefore, $\forall v_1, v_2 \in \mathbb{R}^d$, we have

$$|g_l(v_1, y) - g_l(v_2, y)|$$
$$= |y(w^\top v_2 - w^\top v_1) + \ln\frac{1 + e^{w^\top v_1}}{1 + e^{w^\top v_2}}|$$
$$\le |w^\top v_2 - w^\top v_1| + |\ln\frac{1 + e^{w^\top v_1}}{1 + e^{w^\top v_2}}|$$
$$\le 2|w^\top v_2 - w^\top v_1| \quad (\ln(1 + e^x) \text{ is } 1 - \text{Lipschitz})$$
$$\le 2\theta\|v_2 - v_1\|_2 \quad (\text{Schwarz and } \|w\|_2 \le \|w\|_1).$$

Similarly, since $l(w^\top v, y) = -y\ln(\sigma(w^\top v)) - (1 - y)\ln(1 - \sigma(w^\top v))$, for any $y \in \{0, 1\}$ we can obtain

$$|l(w_1^\top v, y) - l(w_2^\top v, y)| = |y(w_1^\top v - w_2^\top v) + \ln\frac{1 + e^{w_2^\top v}}{1 + e^{w_1^\top v}}| \le 2|w_1^\top v - w_2^\top v|. \quad \blacksquare$$

***Proof of Claim 3 in the main paper.***
For any fixed $y \in [0, 1]$, for any $v_1, v_2 \in \mathbb{R}^d$, notice that both $g(v_1)$ and $g(v_2)$ also lie into the interval $[0, 1]$, then

$$|l(g(v_1), y) - l(g(v_2), y)| = |(g(v_1) - y)^2 - (g(v_2) - y)^2| \le 2|g(v_1) - g(v_2)|. \quad \blacksquare$$

B.5    PROOF OF WHEN THE ENVIRONMENT IS ALMOST $\Pi$-RELATED?

To obtain a complete proof of Theorem 6 in the main paper, we first give an important lemma that states the existence of the almost isomorphism between a complete separable metric space and the unit interval $[0,1]$. We also introduce the formal definitions of *metric Boolean algebra* and *metric Boolean isomorphism* (Bogachev, 2007).

**Theorem 7** *(Bogachev, 2007) Let $(Z, \mu)$ be a complete separable metric space with a Borel probability measure $\mu$. Then $(Z, \mu)$ is almost isomorphic to the space $([0,1], \nu)$, where $\nu$ is some Borel probability measure. If $\mu$ is an atomless measure, then one can take for $\nu$ Lebesgue measure.*

**Definition 8** *(Metric Boolean Algebra) Let $(Z, \mathcal{B}, \mu)$ be a measure space with a finite nonnegative measure $\mu$. Let $d(A,B) = \mu(A \triangle B), A, B \in \mathcal{B}$. The function $d$ is called the Frechet-Nikodym metric and we can introduce the following equivalence relation on $\mathcal{B}$: $A \sim B$ if $d(A,B) = 0$. Then the metric space $(\mathcal{B}/\mu, d)$ is called the metric Boolean algebra, or measure algebra, often denoted by $E_u$.*

Further, the metric space $(\mathcal{B}/\mu, d)$ is separable,i.e.,contains a countable everywhere dense subset, if and only if the corresponding measure $\mu$ is separable. The separability of $\mu$ is equivalent to the existence of a countable collection of sets $B_n \in \mathcal{B}$ such that $\forall B \in \mathcal{B}$ and $\epsilon > 0$, there exists an integer $n$ with $\mu(B \triangle B_n) \leq \epsilon$. In addition, a metric space $(\mathcal{B}/\mu, d)$ is complete if $\mathcal{B}$ is a $\sigma$-algebra and $\mu$ is countably additive (e.g., $\mu$ is a measure).

**Definition 9** *(Metric Boolean Isomorphism) Two measure algebras $E_{u_1}$ and $E_{u_2}$ generated by measure spaces $(Z_1, \mathcal{B}_1, \mu_1)$ and $(Z_2, \mathcal{B}_2, \mu_2)$ are called isomorphic if there exists a one-to-one mapping $\mathcal{J}$ from $E_{\mu_1}$ onto $E_{\mu_2}$ (called a metric Boolean isomorphism) such that $\mathcal{J}$ preserves the measure, i.e., $\mu_2(\mathcal{J}(A)) = \mu_1(A), \forall A \in E_{\mu_1}$.*

**Theorem 8** *(Bogachev, 2007) Every separable atomless measure algebra is isomorphic to the measure algebra of some interval (e.g., [0,1]) with Lebesgue measure.*

**Definition 10** *(Lebesgue-Rohlin Space) A measure space $(Z, \mathcal{B}, \mu)$ is called a Lebesgue-Rohlin space if it is almost isomorphic to some measure space $(Z', \mathcal{B}', \mu')$ with a countable basis with respect to which $Z'$ is complete.*

**Example 1** *The space $([0,1], \mathcal{B}([0,1]), \lambda)$, where $\lambda$ is Lebesgue measure, has a countable basis with respect to which it is complete.*

**Theorem 9** *(von Neumann, 1932) Let $(Z_1, \mathcal{B}_1, \mu_1)$ and $(Z_2, \mathcal{B}_2, \mu_2)$ be Lebesgue-Rohlin spaces with probability measures. If the corresponding measure algebra $E_{\mu_1}$ and $E_{\mu_2}$ are isomorphic in the sense of Definition 9, then there exists an almost isomorphism between these spaces. In particular, this is the case if both measures are atomless.*

***Proof of Theorem 6 in the main paper.***
The proof contains 3 main steps: (1) Since $Z$ is the complete separable metric space and the probability measures $P_i, P_j$ are atomless, then from Theorem 7, $(Z, \mathcal{B}_i, P_i)$ and $(Z, \mathcal{B}_j, P_j)$ are both almost isomorphic to the measure space $([0,1], \mathcal{B}([0,1]), \lambda)$ where $\lambda$ is the Lebesgue measure. From Definition 10 and Example 1, $(Z, \mathcal{B}_i, P_i)$ and $(Z, \mathcal{B}_j, P_j)$ are Lebesgue-Rohlin spaces. (2) From Example 1, the complete measure space $([0,1] \backslash M, \mathcal{B}([0,1])_\lambda, \lambda)$ (w.r.t. the measure $\lambda$ and $\lambda(M) = 0$) has a countable basis $\{B_n\}_{n=1}^\infty \subset \mathcal{B}([0,1])_\lambda$. The almost isomorphism $\pi_i$ from $([0,1], \mathcal{B}([0,1]), \lambda)$ into $(Z, \mathcal{B}_i, P_i)$ can induce a countable basis $\{\pi_i(B_n)\}_{n=1}^\infty \subset \mathcal{B}_i$, which guarantees the separability of the measure algebras $E_{P_i}$. The separability of the measure algebras $E_{P_j}$ can be guaranteed in the same way. Then from Theorem 8, the measure algebras $E_{P_i}$ and $E_{P_j}$, generated by measure spaces $(Z, \mathcal{B}_i, P_i)$ and $(Z, \mathcal{B}_j, P_j)$ , are both isomorphic to the measure algebra of the interval [0,1] with Lebesgue measure. Therefore, the two measure algebra $E_{P_i}$ and $E_{P_j}$ are isomorphic (since isomorphism is an equivalence relation). (3) Combining (1) and (2), and recalling Theorem 9, the two measure spaces $(Z, \mathcal{B}_i, P_i)$ and $(Z, \mathcal{B}_j, P_j)$ are almost isomorphic. So $P_i$ and $P_j$ are almost $\Pi$-related in the sense of Definition 3. ∎

## C  MORE DETAILED COMPARISONS WITH RELATED WORKS

### C.1  DETAILED COMPARISON WITH GENERALIZATION BOUNDS OF (BAXTER, 2000)

This paper can be considered as the extension of the meta learning theoretical work in (Baxter, 2000), by further exploring the task relatedness for the environment. In this section, we detail our improvements over this pioneering work. First, we introduce a new notation called $\mathbb{H}^*$. For any hypothesis space $\mathcal{H} \in \mathbb{H}$, define $\mathcal{H}^* : \mathcal{P} \to [0, 1]$ by $\mathcal{H}^*(P) = \inf_{h \in \mathcal{H}} er_P(h)$, and for any hypothesis space family $\mathbb{H}$, define $\mathbb{H}^* = \{\mathcal{H}^* : \mathcal{H} \in \mathbb{H}\}$. Although Baxter does not give the explicit PAC-style learning bound in his main paper, its bound on $|\hat{er}_{\mathbf{z}}(\mathcal{H}) - er_Q(\mathcal{H})|$ can be expressed as:

$$\sqrt{\frac{64}{mn} \ln \frac{8\mathcal{C}(\epsilon/8, \mathbb{H}_l^n)}{\delta}} + \sqrt{\frac{64}{n} \ln \frac{8\mathcal{C}(\epsilon/8, \mathbb{H}^*)}{\delta}}, \tag{3}$$

where $\mathcal{C}(\epsilon, \mathbb{H}_l^n) = \sup_{\mathbf{P}} \mathcal{N}(\epsilon, \mathbb{H}_l^n, d_{\mathbf{P}}), \mathcal{C}(\epsilon, \mathbb{H}^*) = \sup_Q \mathcal{N}(\epsilon, \mathbb{H}^*, d_Q)$ (see its Theorem 4). Our improvements can be summarized in the following three aspects:

**(1)** We exploit the proposed task relatedness to reduce the complexity $\mathcal{C}(\epsilon, \mathbb{H}^*)$ in Eq. (3) to zero in Theorems 3. In Theorem 6, we further show the rationality of our task relatedness assumption when the sample space $Z$ is a complete separable metric space. Given that (Baxter, 2000) also assumes $Z$ to be a separable metric space to obtain theoretical results, our derived meta learning bound in Theorem 3, albeit depending on a slightly stronger assumption, is a non-trivial enhancement. Nevertheless, we admit that our results also rely on the closure property of the function class $\mathcal{H}_l$.

**(2)** Our covering number results (e.g., Theorem 4) are based on the metric defined w.r.t. the empirical measure, instead of the abstract measure in Baxter's work (e.g., Theorem 6 in (Baxter, 2000)). Therefore it is more suitable to combine our results with the recent theoretical results of covering number bounds for deep neural networks in single task learning.

**(3)** When bounding the covering number (or say capacity) of the neural network $\mathcal{C}(\epsilon_2, \mathcal{F})$ (i.e., Theorem 8 in (Baxter, 2000)), Baxter uses traditional Pseudo-dimension indicator which is developed by (Haussler, 1992), resulting in a parameter-count-based bounds for meta learning with neural network. However, this is not suitable for the analysis of modern overparameterized deep networks. In this paper, we introduce a new complexity indicator, spectrally-normalized bound for deep neural network (Bartlett et al., 2017), into the meta learning framework. The obtained bounds for meta learning with deep network for classification and regression problems in Claims 1-3 are all independent of the size of total parameters, outperforming the results in Theorem 8 of (Baxter, 2000).

**Remark 2** (*The three main technical difficulties when applying spectrally-normalized margin bounds for meta-learning with deep neural networks*)
Now, we give more explanations about the three main technical difficulties when applying spectrally-normalized margin bounds (Bartlett et al., 2017) for meta-learning with deep neural networks.
**(i)** We need to remove the second covering number complexity (i.e. the covering number complexity of $\mathcal{H}^*$) in the meta-learning bound in Eq. (3) in Section C.1 (i.e. the original bound in Theorem 4 of (Baxter, 2000)). Since such covering number complexity is defined (and can only be defined) based on the distance with respect to the abstract measure (instead of the empirical measure, see Definitions 3-4 in (Baxter, 2000)), we cannot use any modern theoretical results in deep learning (e.g., the spectral norm of the neural networks in (Bartlett et al., 2017), the compression bound in (Arora et al., 2018) and the ALL-layer margin in (Wei & Ma, 2020)) but the traditional VC-dimension to bound such covering number complexity, hence leading to the vacuous parameter-count-based bounds for deep neural networks. The aforementioned challenge motivated us to propose the $\Pi$-relatedness notation to measure the similarity between different tasks and finally removed the second covering number complexity in Eq. (3) to obtain our main generalization bound in Theorem 3 that can fully utilize the whole $n * m$ training samples.
**(ii)** For the first covering number complexity in Eq. (3) (which is still defined based on the distance w.r.t. the abstract measure in Baxter's original paper), we still need to transform it into the complexity defined on the distance w.r.t. the empirical measure with our proposed techniques in Propositions 1-2 (also see our Definition 6), hence we can extend our generalization bound in Theorem 3 to the representation setting and make our meta-learning bound with representation learning (i.e. our Theorem 4) much easier to be combined with modern theoretical results (Bartlett et al., 2017; Arora et al., 2018; Wei & Ma, 2020) for deep neural network in single-task learning.
**(iii)** The last difficulty is to connect the covering number complexity in meta-learning setting (e.g.

see our Definition 6, w.r.t. the empirical measure) with the spectrally-normalized based covering number complexity from Bartlett et al. (2017) (i.e. Lemmas 4-5 in our Appendix B.4). Such difficulty is overcome by using the Lipschitness property of neural networks and the theoretical properties of covering number (see the proof of Theorem 5 in Appendix B.4).

## C.2 Detailed Comparison with Task Relatedness Notion of (Ben-David & Schuller, 2003)

In this section, we detail more distinctions between our proposed *almost* $\Pi$-*relatedness* notation and the task relatedness notion defined in (Ben-David & Schuller, 2003). We first recall this previous concept in (Ben-David & Schuller, 2003, Definition 2.1).

**Definition 11** *(Ben-David & Schuller, 2003) Let $\mathcal{F}$ be a set of transformations $f : X \rightarrow X$ , and let $P_1, P_2$ be probability distributions over $X \times \{0,1\}$. We say that $P_1, P_2$ are $\mathcal{F}$-related distributions if there exists some $f \in \mathcal{F}$ such that for any $T \subset X \times \{0,1\}$, $T$ is $P_1$-measurable iff $f[T] = \{(f(x), b) | (x, b) \in T\}$ is $P_2$-measurable and $P_1(T) = P_2(f[T])$.*

**Remark 3** *(**The main differences between our task relatedness notation and that of (Ben-David & Schuller, 2003)**)*
Note that in Definition 11, the condition $P_1(T) = P_2(f[T])$ means that $P_1(T) = P_2 \circ (f \times I)[T]$ for all $T \subset X \times \{0,1\}$. Therefore $P_1 = P_2 \circ (f \times I)$ and hence $f \times I$ is a (bijective) measure-preserving transformation on the product space $X \times \{0,1\}$. Recall the definition of our proposed task relatedness concept in Definition 3 in the main paper. It's not difficult to see that there are two main differences between these two task-relatedness notions: **(i)** The bijective transformation $f$ in Definition 11 of David's work is defined between the input space $X$ and the input space $X$. While in our work, the bijective transformation $\pi$ in Definition 3 of the main paper is defined between the sample space $Z(= X \times Y)$ and the sample space $Z$. Note that the bijective transformation $f$ in Definition 11 can be viewed as the bijective transformation $f \times I$ on the product space $X \times Y$, where $I$ is the identity map on the output space $Y$. Therefore the transformation given by (Ben-David & Schuller, 2003) can be considered as a special case of our proposed task-relatedness concept. **(ii)** Furthermore, the bijective transformation $f$ given in Definition 11 is a function map imposed on the *whole* input space $X$, whereas our proposed bijective transformation $\pi$ is imposed on the *almost whole* sample space $Z$ excluding a measure-zero set. In other words, our proposed $\pi$ is a bijective map between $Z \backslash N$ and $Z \backslash N_1$, where $N, N_1$ are $P$-measure zero and $P_1$-measure zero sets, respectively. Such design will allow more flexibility of the choice of the bijective map $\pi$, and can help us theoretically guarantee the existence of the *almost* bijective function on the whole sample space $Z$ (see more explanations in Section B.5). In contrast, (Ben-David & Schuller, 2003) does not provide a rigorous demonstration of the existence of its defined bijective transformation $f$. Actually, it is not easy to find such bijective transformation $f$ on the input space $X$, which meanwhile satisfies the condition that the bijective transformation $f \times I$ is a measure-preserving bijective on the product space $X \times Y$. In this sense, our work can be considered as the extension version of that in (Ben-David & Schuller, 2003).

As for our proposed task-relatedness concept, the explicit form of the bijective transformation $\pi$ in Theorem 6 of main paper can be found in (Bogachev, 2007, Theorem 6.5.7). In that result, Bogachev takes $\pi$ as the composition of functions $\sum_{n=1}^{\infty} I_{E_n}$ and $(\sum_{n=1}^{\infty} I_{B_n})^{-1}$, where $E_n$ belongs to the $\sigma$-algebra $\mathcal{B}_\mu$ over the sample space $Z$ associated with the measure $\mu$, $B_n$ belongs to the $\sigma$-algebra $\mathcal{B}_\nu$ over the sample space $Z$ associated with the measure $\nu$, $I$ is the indicator function from the $\sigma$-algebra over $\mathcal{Z}$ to the interval $[0, 1]$. A simpler (but not rigorous) example just for intuitive comprehension can be found in the regression problem, where the sample space is $Z = \mathbb{R}^{d+1} = X \times Y = \mathbb{R}^d \times \mathbb{R}$, the focused measure is Lebesgue measure and the bijective function $\pi : \pi(\vec{z}) = \vec{z} + \vec{a}, \vec{a} \in Z$ is a translation over high-dimensional Euclidean space.

