# OpenReview forum: "Task Relatedness-Based Generalization Bounds for Meta Learning"
_ICLR.cc/2022/Conference — ICLR 2022 Spotlight_

### Official Review · Reviewer_VEYq · 2021-11-02

**Correctness:** 4
**Technical Novelty And Significance:** 4
**Empirical Novelty And Significance:** Not applicable
**Recommendation:** 8
**Confidence:** 3

**Main Review:**

Significance: The paper takes a meaningful and important step towards developing meta-learning theory. The paper addresses one of the key issues in Baxter (2000) of not being able to fully utilize m*n training samples. This is done by imposing the task-relatedness assumption on the task environment. The paper later provides upper bound on the covering number, particularly in the representation learning with neural network setting.

Novelty: The paper follows Baxter’s framework, and the notion of task-relatedness is also previously studied in Ben-David & Schuller (2003), albeit with some differences (highlighted by the authors in the appendix). This distinguishes itself from recent meta-learning theory work, e.g., Du et al. (2020), and Tripuraneni et al. (2020), that study meta-learning for a fixed target task using the notion of task diversity/similarity.

Presentation: The paper writing and presentation is great. The related work is discussed in good detail. The connection between the proposed theory and existing literature are highlighted and discussed.

Weakness: From what I understand, the proof of the key results uses techniques and results from existing works. The PAC-style generalization bounds (Theorem 1, 2, 3) use results from Baxter (2000). The covering number bounds with deep neural networks (Theorem 5) use results from Bartlett (2017). It would be nice if the authors can highlight some challenges/difficulties in adapting existing results.


**Summary Of The Paper:**

The paper presents generalization bounds for meta-learning (Corollary 1) that obtain O(1/sqrt{m*n} * log{covering number}) convergence rate. Unlike previous work, the proposed bound can utilize all m*n training samples (n tasks, m samples per task). In a special branch of meta learning that involves representation learning with neural networks, i.e., the algorithm learns a common embedding that is shared across tasks, the paper establishes spectrally-normalized bounds for both classification and regression problems.

The paper relies on two critical assumptions: relatedness between tasks and closure property of the hypothesis space. The paper shows that the task-relatedness assumption is satisfied if the sample space is a complete and separable metric space. The paper also briefly discusses the conditions for which the closure property assumption holds.


**Summary Of The Review:**

The paper makes an important and meaningful contribution to the theory of meta-learning. The paper is well-written, and the ideas presented are sufficiently novel. I think the paper would make for a good submission in ICLR 2022.

---

### Official Review · Reviewer_bHHr · 2021-11-02

**Correctness:** 3
**Technical Novelty And Significance:** 3
**Empirical Novelty And Significance:** Not applicable
**Recommendation:** 8
**Confidence:** 3

**Main Review:**

Strengths.
As far as I know, the bounds are novel, require some technical skills, and give novel interesting insights for meta-learning.
The bounds cover different settings (regression, binary/multi-task).
The spectrally normalized bounds are interesting and give more practical insight on the performances of some models.

Weaknesses
The practical use of the bounds is not really discussed, how far can we derived new algorithms from these bounds.
The elements that can be easily estimated and those that may not could  be better discussed.
The information provided by the bounds could be better discussed as well, in particular it not clear if they are informative, maybe giving some situations where the bounds are not loose could be interesting.

About the existence of the $\Pi$-relatedness notion: the authors provide some mathematical frameworks for justifying the existence the $\Pi$-relatedness notion, this is interesting but remains abstract in a sense. In practice, one can wonder if the restriction on the one-to-one mapping is realistic, but no algorithm/method is given for estimating it. I am wondering if optimal transport, using Kantorovich-based relaxations, can provide a setting where some transformations always exist (up to a barycenter mapping), then maybe notions of Wasserstein distances should be considered here.

A paper that can be related:
Wang H. et al. 2021. Bridging Multi-Task Learning and Meta-Learning: Towards Efficient Training and Effective Adaptation. In ICML 2021

--
After rebuttal
--
I'm ok with the answers provided by the authors, I keep my score.
About OT: my point was to use a kind of "one-to-many" mapping provided by the Kantorovich relaxation, but I understand that this not easy.

**Summary Of The Paper:**

The paper presents novel generalization bounds for meta-learning based on a notion of task-relatedness that allows one to compare two tasks by notably allowing a mapping only in subregions where the similarity can be measured in a sense. The theoretical results a covering number bound, a covering number meta-learning bound with representation learning, and spectrally-normalized bounds for meta-learning with neural networks adapted to binary, multi class and regression. The contribution is theoretical.

**Summary Of The Review:**

The paper presents novel and interesting generalization bounds for meta-learning, they offer new insightful results, in particular with spectrally-normalized bounds that are adapted to learn with neural networks in different settings (binary, multi-class, regression).
The practical implication of these results is not really discussed and the perspectives offered by these results for designing new algorithms are not sufficiently developed.

---

### Official Review · Reviewer_7ER7 · 2021-11-03

**Correctness:** 3
**Technical Novelty And Significance:** 4
**Empirical Novelty And Significance:** Not applicable
**Recommendation:** 8
**Confidence:** 2

**Main Review:**

The paper is well-written and comprehensive. Although the paper employed many notations, they are intuitive and easy to follow. Overall I think this is a good paper.

Here are a few specific questions:
1. Can two multi-class classification tasks with different numbers of classes be almost PI-related? Can two tasks with different number of features be almost PI-related? It is unclear from the paper.
2. It is unclear to me how we should separate the embedding function f with the prediction function g. Presumably both the embedding and prediction layers are multi-layer networks, and there are various ways to separate the whole network into an f function and a g function. Could the authors elaborate?

**Summary Of The Paper:**

In this paper, the authors proposed two key concepts: almost PI-Related tasks and almost PI-related environment. Intuitively, if two tasks are PI-related, then if a function space H contains a good solution to task A, it should also contain a good solution to task B.

With these two concepts, the authors went forward and proved that in an almost PI-related environment, the generalization bound has a convergence rate of O(1/\sqrt(mn)). In other words, when tasks are similar, then meta-learning algorithms should be able to utilize all data points across all tasks.

**Summary Of The Review:**

This is a well-written paper with vast implications in the meta-learning society. The almost Pi related concepts are useful and easy to understand.

---

### Official Review · Reviewer_fpV1 · 2021-11-05

**Correctness:** 4
**Technical Novelty And Significance:** 3
**Empirical Novelty And Significance:** Not applicable
**Recommendation:** 8
**Confidence:** 1

**Main Review:**

With the big caveat that I am not knowledgeable on recent theoretical work for meta-learning to provide a very educated critique of this paper, I felt that proposed $\Pi$-relatedness notion and corresponding PAC generalization bound involving m*n samples is novel enough to make this paper interesting to the ICLR community. I have not checked the proofs in the Appendix.

**Summary Of The Paper:**

The paper:
1) proposes a new notion of task-relatedness for meta-learning termed "almost $\Pi$-relatedness" that assumes isomorphisms between two tasks in the environment. A PAC-style generalization bound of $\mathcal{O}(\sqrt{\frac{C}{mn}} +  \sqrt{\frac{\ln(1/\delta)}{mn}})$ is shown, improving upon Baxter et. al's.
2) For meta learning that involves representation learning, they bound $C$ with two covering numbers that are both defined over a single task, making their results suitable to be combined with recent works of deep neural network in the single task learning.
3) They demonstrate that any two tasks in the environment are almost $\Pi$-related if the focused sampled space is a complete separable metric space.

**Summary Of The Review:**

Unfortunately, I am not knowledgeable on recent theoretical work for meta-learning to provide a very educated critique of this paper. The proposed $\Pi$-relatedness notion seems like a novel contribution and the PAC generalization bound involving mn samples seems stronger than existing work, so for that reason I'd lean to accept.

---

### Decision · Program_Chairs · 2022-01-20

**Decision:**

Accept (Spotlight)

**Comment:**

This paper provides generalization bounds for meta-learning based on a notion of task-relatedness. The result is natural and interesting--intuitively, when tasks are similar, then meta-learning algorithms should be able to utilize all data points across all tasks. The theoretical contribution is novel, and the results also provide more practical insight into the performances of some models.